# Tracking Changes in the Area, Thickness, and Volume of the Thwaites Tabular Iceberg 'B30' using Satellite Altimetry and Imagery

Anne Braakmann-Folgmann[1], Andrew Shepherd[1], Andy Ridout[2]

[1] Centre for Polar Observation and Modelling (CPOM), University of Leeds, Leeds, LS2 9JT, UK
[2] Centre for Polar Observation and Modelling (CPOM), University College London, London, UK

*Correspondence to*: Anne Braakmann-Folgmann (eeabr@leeds.ac.uk)

**Abstract.** Icebergs account for half of all ice loss from Antarctica and, once released, present a hazard to maritime operations. Their melting leads to a redistribution of cold fresh water around the Southern Ocean which, in turn, influences water circulation, promotes sea ice formation, and fosters primary production. In this study, we combine CryoSat-2 satellite altimetry with MODIS and Sentinel-1 satellite imagery and meteorological data to track changes in the area, freeboard, thickness, and volume of the B30 tabular iceberg between 2012 and 2018. We track the iceberg elevation when it was attached to Thwaites Glacier and on a further 106 occasions after it calved using Level 1b CryoSat data, which ensures that measurements recorded in different acquisition modes and within different geographical zones are consistently processed. From these data, we mapped the iceberg's freeboard and estimated its thickness taking snowfall and changes in snow and ice density into account. We compute changes in freeboard and thickness relative to the initial average for each overpass and compare these to estimates from precisely located tracks using the satellite imagery. This comparison shows good agreement (correlation coefficient 0.87), and suggests that colocation reduces the freeboard uncertainty by 1.6 m. We also demonstrate that the snow layer has a significant impact on iceberg thickness change. Changes in the iceberg area are measured by tracing its perimeter and we show that alternative estimates based on arc lengths recorded in satellite altimetry profiles and on measurements of the semi-major and semi-minor axes also capture the trend, though with a 48 % overestimate and a 15 % underestimate, respectively. Since it calved, the area of B30 has decreased from $1500 \pm 60$ to $426 \pm 27$ km$^2$, its mean freeboard has fallen from $49.0 \pm 4.6$ to $38.8 \pm 2.2$ m, and its mean thickness has reduced from $315 \pm 36$ to $198 \pm 14$ m. The combined loss amounts to an $80 \pm 16$ % reduction in volume, two thirds ($69 \pm 14$ %) of which is due to fragmentation and the remainder ($31 \pm 11$ %) is due to basal melting.

## 1 Introduction

Iceberg calving accounts for roughly half of all ice loss from Antarctica (Depoorter et al., 2013; Rignot et al., 2013). At any time, about 50-90 large tabular icebergs are tracked in the Southern Ocean containing 7 000 to 17 000 km³ of ice in total (Tournadre et al., 2015). For maritime operators it is essential to know the location of icebergs in order to reduce the

risk of collision (Bigg et al., 2018; Eik and Gudmestad, 2010; Power et al., 2001). The thickness of an iceberg determines if and where it will ground on the seabed, which has implications for maritime operations as well as for marine geophysics. Iceberg thickness also influences a wide range of physical and biological interactions with the Antarctic environment. Grounded icebergs can, for example, alter the local ocean circulation (Grosfeld et al., 2001; Robinson and Williams, 2012), influence melting of the adjacent ice shelves (Robinson and Williams, 2012), and prevent local sea ice from breaking up

(Nøst and Østerhus, 2013; Remy et al., 2008). This, in turn, can impact the local primary production (Arrigo et al., 2002; Remy et al., 2008) and pose an obstacle to penguin colonies on their way to their feeding grounds (Kooyman et al., 2007). Temporarily grounded icebergs leave plough marks on the sea floor which can be an important geological record (Wise et al., 2017) but also impact on marine benthic communities (Barnes, 2017; Gutt, 2001). Therefore, iceberg thickness is an important parameter.

Changes in iceberg thickness are also important, because they control the quantity of cold fresh water and terrigenous nutrients released into the ocean as icebergs melt (Gladstone et al., 2001; Silva et al., 2006). The release of relatively cold fresh water facilitates sea ice growth (Bintanja et al., 2015; Merino et al., 2016), immediately lowers the sea surface temperature (Merino et al., 2016), and has been found to even influence ocean water down to 1500 m depth (Helly et al., 2011) as well as lead to upwelling of deep ocean properties (Jenkins, 1999). In terms of nutrients, icebergs have shown to

be the main source of iron in the Southern Ocean (Laufkötter et al., 2018; Raiswell et al., 2016; Wu and Hou, 2017) and therefore foster primary production in the proximity of icebergs (Biddle et al., 2015; Duprat et al., 2016; Helly et al., 2011), which in turn increases the abundance of krill and seabirds (Joiris, 2018; Smith et al., 2007) around icebergs. Furthermore, a range of studies have demonstrated that including more realistic iceberg distributions, trajectories, and volumes in climate models leads to a redistribution of fresh water and heat flux, which agrees better with observations than models that only

include small icebergs or that treat iceberg discharge as coastal runoff (Jongma et al., 2009; Martin and Adcroft, 2010; Rackow et al., 2013; Schloesser et al., 2019). To investigate each of these processes and interrelations, knowledge of iceberg thickness and volume and their change over time is required (England et al., 2020; Merino et al., 2016). Moreover, monitoring iceberg melting also presents an opportunity to gain insights into the response of glacial ice to warmer environmental conditions, which may develop at ice shelf barriers in the future (Scambos et al., 2008; Shepherd et al.,

55     2019).

The first detailed studies on iceberg melting were performed in the 1970's and 1980's, and were mainly based on laboratory experiments or ship-based observations (Hamley and Budd, 1986; Huppert and Josberger, 1980; Neshyba and Josberger, 1980; Russell-Head, 1980). These studies found that iceberg melting, to first order, is proportional to the water temperature and that for large icebergs breakage dominates over melting. More recently, Silva et al. (2006) and Jansen et al. (2007) modelled melting of giant icebergs and the associated fresh water fluxes. The latter found that melting does not only depend on ocean temperature but also on iceberg drift speed and the surrounding ocean currents. Scambos et al. (2008) installed a range of measurement tools including a GPS receiver, a pre-marked accumulation mast and buried bamboo poles observed with a camera on a large Antarctic iceberg to monitor melting. They differentiate between three kinds of mass loss: rift calving, edge wasting, and rapid disintegration. While rift calving can occur at any time within the iceberg life cycle along pre-existing fractures, edge wasting is only observed outside the sea ice edge. Rapid disintegration is caused by surface melting and the formation of surface lakes.

The advent of satellite remote sensing greatly increased our capability to study icebergs – especially the largest ones. A wide range of studies have employed repeat satellite imagery to track changes in iceberg area (Bouhier et al., 2018; Budge and Long, 2018; Collares et al., 2018; Han et al., 2019; Li et al., 2018; Mazur et al., 2019; Scambos et al., 2008). The most common approach to measure iceberg thickness is using satellite altimeter measurements of their freeboard, which began in the late 1980's (McIntyre and Cudlip, 1987). Since then, a range of studies have employed laser and radar altimetry to study freeboard change of large tabular icebergs: Jansen et al. (2007) studied the A-38B iceberg in the Weddell and Scotia Sea with a combination of laser and radar altimetry, and Scambos et al. (2008) also included three Ice, Cloud and land Elevation Satellite (ICESat) overpasses over the A22A iceberg to derive its thickness change. Both studies make use of satellite imagery to colocate the altimetry tracks and to compare similar areas in terms of freeboard change. In contrast, Tournadre et al. (2015) employed altimetry measurements from Envisat, Jason1, and Jason2 to analyse freeboard change of the C19A iceberg without any colocation. Bouhier et al. (2018) analysed thickness changes of the B17A and C19A icebergs in open water using altimetry data without colocation. Li et al. (2018) calculated freeboard change of the C28A and C28B icebergs for two years at the intersections of CryoSat-2 overpasses, and Han et al. (2019) also used intersecting CryoSat-2 tracks to calculate freeboard change of the A68 iceberg in the Weddell Sea. When thickness and area changes are combined, it is possible to detect changes in iceberg volume (Bouhier et al., 2018; Han et al., 2019; Tournadre et al., 2012). However, studies to date have been limited to selected icebergs, have focussed on the Weddell Sea, and have employed a variety of approaches to account for the irregular sampling of altimetry tracks including manual colocation of entire tracks relative to the initial surface (Jansen et al., 2007), colocation of intersecting tracks (Han et al., 2019; Li et al., 2018), and with no colocation at all (Bouhier et al., 2018; Tournadre et al., 2015). For smaller icebergs satellite stereo photogrammetry (Enderlin and Hamilton, 2014; Sulak et al., 2017) and interferometry (Dammann et al., 2019) have been

employed to measure iceberg thickness and volume as an alternative approach, though in our experience both methods are labour intensive.

In this study, we quantify changes in the area, freeboard, thickness, and volume of the giant tabular B30 iceberg, which has been adrift in the Southern Ocean since it calved from the Thwaites Glacier 8.5 years ago (Budge and Long, 2018; Fig. 1). The long life-cycle and large drift of the B30 iceberg result in a relatively high number of observations, enabling a detailed study of its evolution. This is also one of the first studies to investigate iceberg thinning in the Southern Ocean around Marie Byrd Land. We assess the agreement between estimates of freeboard change determined relative to the average initial surface and using precise colocation with the aid of near-coincident satellite imagery. Moreover, we develop a methodology to account for snowfall and the evolutions of snow and ice density and examine the influence of snow on the iceberg thickness calculation. The next chapter introduces the remote sensing data used in this study and explains our methodology; the third chapter presents our results on iceberg area, freeboard, thickness, and volume change in turn and discusses our findings. We close with conclusions and a brief outlook in chapter four.

## 2 Data and methods

To chart the iceberg area change over time we delineate its extent in a sequence of Moderate Resolution Imaging Spectroradiometer (MODIS) optical satellite imagery and Sentinel 1 synthetic aperture radar (SAR) satellite imagery. We then use CryoSat-2 satellite radar altimetry to determine changes in the iceberg freeboard and thickness, assuming that it is floating in hydrostatic equilibrium, and making use of the iceberg orientation relative to its initial position using near-coincident satellite imagery on some occasions. We account for snow accumulation and model variations in snow and ice density when converting iceberg freeboard to thickness. Finally, we combine both data sets to estimate the iceberg's volume change over time.

### 2.1 Iceberg location

We use daily archived iceberg positions from the Antarctic Iceberg Tracking (AIT) database version 3.0 provided by the Brigham Young University (Budge and Long, 2018) as a baseline estimate of the B30 iceberg location since it calved in 2012 (Fig. 1). The AIT database makes use of coarse-resolution passive microwave scatterometer imagery in which icebergs are manually detected and the central position is recorded daily (Stuart and Long, 2011). It includes icebergs longer than 6 km adrift in the Southern Ocean between 1987 and 2019, augmented with estimates of position and the semi minor and major axes lengths of icebergs longer than 18.5 km that are tracked operationally by the U.S. National Ice Center (NIC) using a combination of visible, infrared, and SAR imagery.

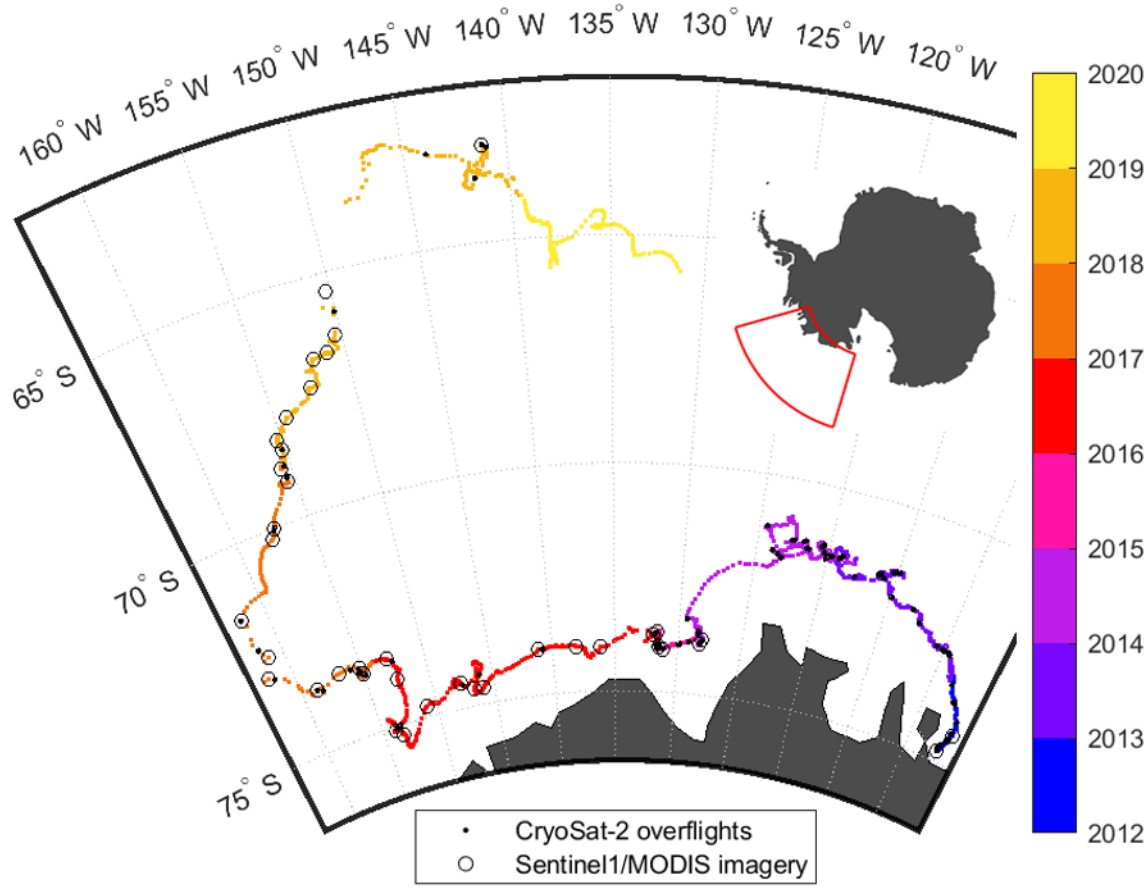

**Figure 1: Trajectory of the B30 iceberg as recorded by the Antarctic Iceberg Tracking Database (Budge and Long, 2018): After calving from the Thwaites Ice Shelf in 2012, it followed the coastal current westwards, started drifting north in 2017 and eventually disintegrated in 2019. Black dots mark the positions where CryoSat-2 overflights over the iceberg are available, circles depict the positions of the MODIS and Sentinel 1 images used in this study**

## 2.2 Initial iceberg shape, size and calving position

To determine the initial shape, size, and calving position of B30, we use MODIS images acquired before and after the calving event to identify which section of the Thwaites Ice Shelf calved to form the iceberg. MODIS is an instrument on the Terra and Aqua satellites by NASA launched on 18[th] December 1999 and 4[th] May 2002, respectively. The instrument measures radiance in the visible and infrared range with a spatial resolution of 250 m to 1 km and covers the entire Earth in 1-2 days, though cloud occlusions and the absence of daylight reduce data availability for many applications. For this

study we use bands 1 (red), 4 (green), and 3 (blue) of the MODIS Level 1B calibrated radiances at 500 m resolution (MOD02HKM). As B30 broke off on 24 May 2012 (Budge and Long, 2018) in Antarctic winter, during darkness, the closest useful MODIS imagery is from the preceding autumn and subsequent spring. We use several MODIS images

acquired in the subsequent spring after calving to determine the initial shape, as it is difficult to unambiguously distinguish the berg from clouds and sea ice in a single image. The initial perimeter (Figure 2a, 3a) was then shifted and rotated to fit the situation before calving to identify the part of the Thwaites ice shelf that formed B30 (Fig. 4). The initial area (in plan-view) of the iceberg is 1500 km$^2$ with a long axis of around 59 km (Budge and Long, 2018).

### 2.3 Iceberg area

We employ three approaches to estimate the plan-view iceberg area; (i) manual delineation in sequential satellite imagery scenes, (ii) using measurements of the semi-major and semi-minor axes provided by the NIC and assuming an elliptical shape, and (iii) using measurements of their arc lengths recorded in satellite altimetry and assuming a circular shape. While manual delineation provides the most consistent and accurate area estimate, the axes and arc length approaches are much simpler to implement and can be fully automated.

Our main approach to determine iceberg area is manual delineation using a sequence of 32 Sentinel-1 SAR and 8 MODIS optical images. Sentinel 1A and 1B are companion imaging radar satellites launched by the European Space Agency on 3$^{rd}$ April 2014 and 25$^{th}$ April 2016, respectively. Together, they provide repeat sampling of the Earth's surface every 6 days. For this study, we use Level 1 Ground Range Detected (GRD) data. Depending on availability, both interferometric wide (IW) and extra wide (EW) swath mode are used, but over the open ocean only EW data are acquired. We employ the

Sentinel Application Platform (SNAP) toolbox to apply the orbital and radiometric corrections provided with the imagery. The SAR images were multi-looked with a factor of six to reduce speckle and computation time, leading to a spatial resolution of 240 m. Finally, a terrain correction was applied using the GETASSE30 (Global Earth Topography And Sea Surface Elevation at 30 arc second resolution) digital elevation model. The resulting backscatter values are scaled between their 5$^{th}$ and 95$^{th}$ percentiles. The MODIS optical imagery were required prior to the launch of Sentinel-1A in 2014.

To chart changes in the iceberg area over time, we delimit its outline as a polygon in each subsequent image (Fig. 2, see also Bouhier et al., 2018; Collares et al., 2018; Han et al., 2019). When the iceberg is drifting in open water its outline can be detected automatically using boundary detection techniques (e.g. using matlab's bwboundaries function). However, in the presence of sea ice the iceberg could not be separated using this approach, and so we instead delimit its outline manually on such occasions (Bouhier et al., 2018). If parts of the iceberg are covered by clouds, we again use multiple MODIS

images together, so that different parts of the iceberg are obscured by clouds in each image (e.g. Fig 3l). Also sea ice frozen to the iceberg is easier to distinguish from its colour and texture, when several images are used together (e.g. Fig 3b, c). To

estimate the uncertainty of our delineations, we buffer the polygons by the source imagery pixel width (500 m for MODIS images and 240 m for multi-looked Sentinel 1 images) and calculate the resulting difference in area. This gives a mean relative difference of 3.6 ± 0.9 %.


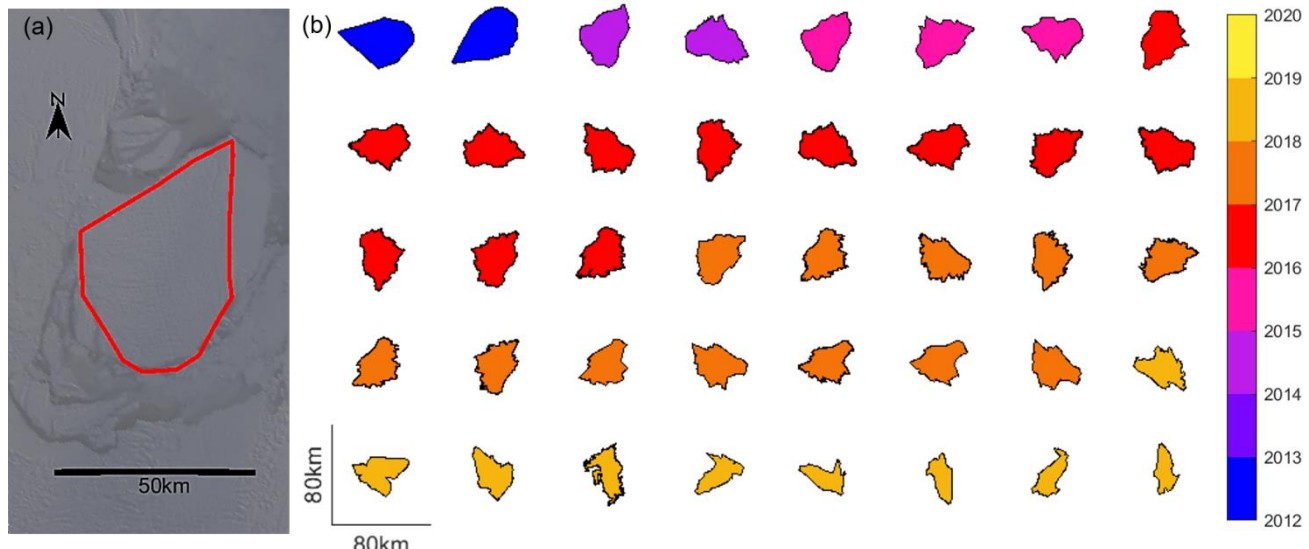

**Figure 2: Outlines of the B30 iceberg derived from satellite imagery. a) Initial shape (red polygon) of the B30 iceberg determined from MODIS images after calving; the background is a MODIS image on 11 September 2012. b) Polygon outlines derived from further MODIS and Sentinel 1 imagery plotted in polar stereographic projection and used to calculate area change of the B30**
**iceberg.**

Our second method of estimating the iceberg area is based on 228 measurements of the semi-major and semi-minor axes lengths. Although iceberg area is most accurately calculated from delineation of their full perimeter in satellite images, the downside of this approach is that it requires a high degree of time-consuming manual interaction and clear imagery. This also makes it less reproducible and subject to individual judgement. We take the size of an ellipse calculated from the semi
major and minor axis provided by the NIC and compare this with our imagery-based iceberg area calculations. The NIC operationally tracks icebergs longer than 18.5 km using a combination of visible, infrared, and SAR imagery. Observations are made weekly, but especially in the early days longer data gaps exist, and not every estimate of semi axes length is based on a new manual observation, but some are just duplicated from the previous observation. Their estimates of semi axes lengths are also rounded to nautical miles (1.852 km), leading to a stepwise evolution of iceberg area with only 8 different
estimates. We base our trend estimate and analysis solely on these 8 estimates, because we are confident that these are unique observations. The uncertainty of this approach is governed by the assumption of an elliptical iceberg shape and the irregular, rounded updates.

Our third and final method of estimating the iceberg area is to make use of 106 CryoSat-2 satellite altimeter overpasses, which are also used to calculate the iceberg's thickness. We record the arc lengths of the iceberg sampled by these tracks
and estimate iceberg area by assuming the iceberg has a circular shape. Depending on the position and relative orientation of the iceberg with respect to each overpass, CryoSat-2 will occasionally sample the long axis but more often a shorter corner. This leads to considerable variations in the area estimates, and in general an underestimation. We employ a ten-point moving mean over time to reduce the variability. The principal uncertainty of this approach is because one-dimensional arc lengths cannot reliably represent a two-dimensional area especially when the shape is evolving and if it is
unknown which part of the shape was sampled.

## 2.4 Iceberg orientation

To track the iceberg shape and rotation in later images relative to its initial orientation, we record the iceberg's orientation in all satellite images that are near-coincident in time with CryoSat-2 overflights (Fig. 3). To orientate the iceberg, we manually identify the coordinates of one corner of the initial iceberg polygon outline at the time of each new overpass and
adjust the rotation angle to align (colocate) all images to a common orientation (Fig.7a-l). This allows us to transform the iceberg coordinates at the time of each image acquisition relative to the equivalent position at the time just before it calved.

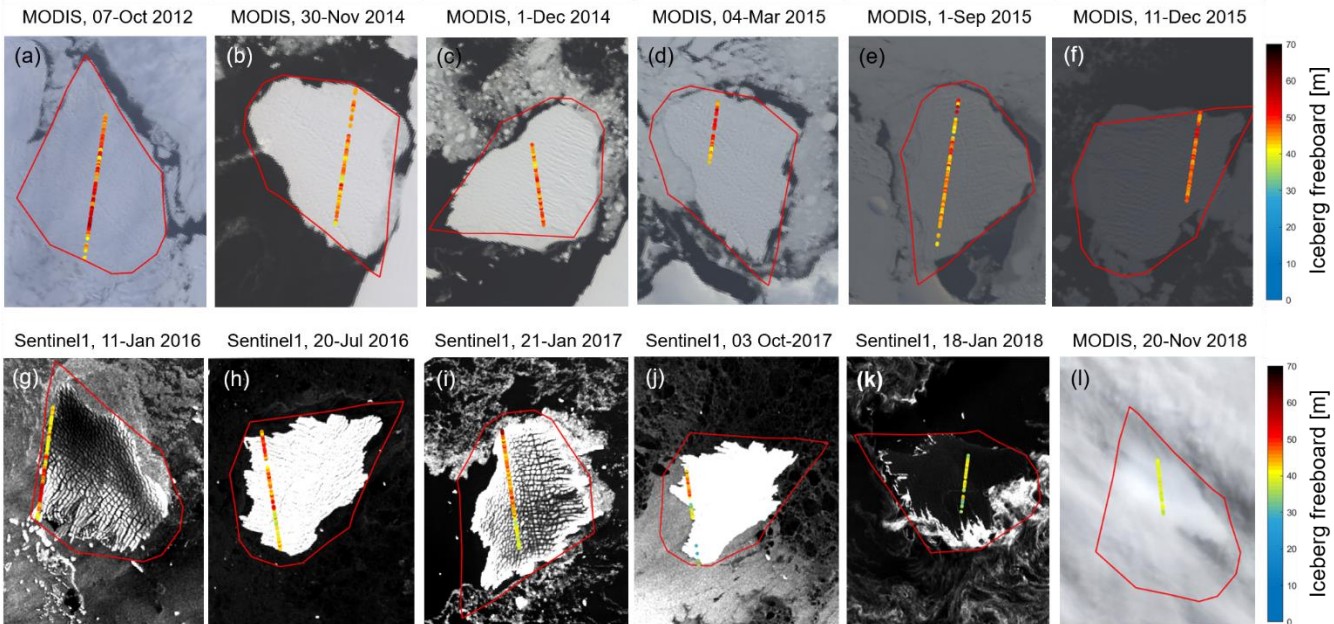

**Figure 3: Satellite imagery with near-coincident CryoSat-2 tracks of iceberg freeboard and the manually transformed initial polygon shape plotted on top. The initial polygons are used to determine the relative position of each new overpass.**

**2.5 Initial iceberg freeboard**

We use CryoSat-2 satellite altimetry to determine freeboard and thickness of the B30 iceberg. CryoSat-2 is a satellite radar altimeter that employs SAR processing to achieve along track resolution of 250 m. It was launched by the European Space Agency on 8 April 2010 in a 369-day repeat period with a 30-day sub cycle. We use Level 1B baseline C data from the CryoSat-2 Science server and apply the Centre for Polar Observation and Modelling sea ice processing system (Tilling et al., 2018) to deduce surface height. For consistency, a common threshold retracker is applied to measurements acquired in both SAR and SAR interferometric mode and over all surface types. Using Level 1B data is important, because the Level 2 products are generated using different retrackers and different biases for different modes and surface types, and so the signals acquired during different parts of the iceberg trajectory are not comparable. Iceberg freeboard is calculated by subtracting the adjacent mean sea surface height from the iceberg surface height.

Although satellite altimeters only sample icebergs along 1-dimensional profiles beneath their ground track while they are drifting, it is possible to build up a detailed 2-dimensional picture of their surface over time prior to calving while their movement is relatively modest. To map the initial freeboard height of B30, we combine all CryoSat-2 tracks recorded within almost 5 months (1 January 2012 to 24 May 2012) before it calved (Fig. 4a). The Thwaites Ice Shelf flows at 3.9 km per year on average (Mouginot et al., 2019), and so we adjust earlier tracks to account for this movement. Because the Thwaites Ice Shelf has a particularly rugged and crevassed surface topography, the point-of-closest-approach (POCA) varies. To make different overpasses more comparable, we remove outliers by deleting freeboard heights greater than 60 m or below 20 m freeboard (Tournadre et al., 2015), and crevasses by deleting freeboard heights falling either below the median minus one standard deviation or below the 5-point moving mean minus the 5-point moving standard deviation. After outlier removal, the mean initial iceberg freeboard is 45.5 m above the adjacent sea level with a wide spread of 8.1 m standard deviation. When crevasses are excluded, the mean freeboard is 49.0 m with a much lower standard deviation of 4.6 m. Because the resulting freeboard measurements are still quite sparse, we average them within 5 km grid cells to obtain a continuous reference surface (Fig. 4). The number and standard deviation of the gridded freeboards give an indication of the variance within each grid cell. The mean standard deviation within each grid cell is 3.3 m, the standard deviation across different grid cells is 3.1 m and the overall standard deviation of all heights within the polygon is 4.6 m. We compare the gridded initial freeboard to measurements from the first CryoSat overpass when the iceberg is adrift, acquired shortly after calving, to check they are consistent, and find a mean difference of -0.4 m. As this value is considerably lower than the iceberg freeboard variability, we conclude that the ice shelf was floating freely prior to calving also, and that the gridded heights are representative of the initial freeboard.

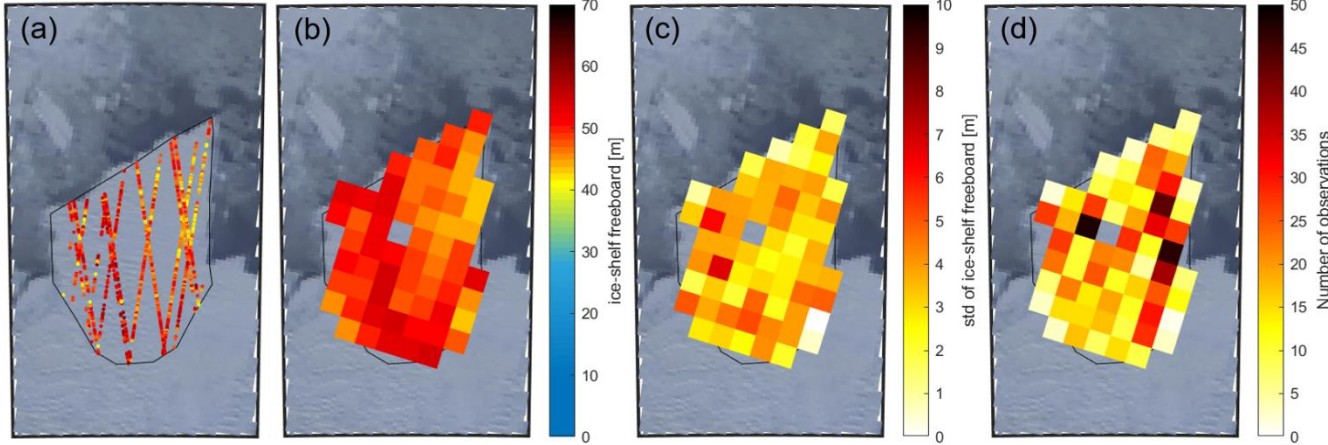

**Figure 4: Initial freeboard heights of the B30 iceberg overlain on a MODIS image on 19 March 2012 (before calving). a) Filtered CryoSat-2 measurements of 145 days before calving, b) Gridded CryoSat-2 data, c) Standard deviation of the gridding, d) Number of measurements per grid cell**

## 2.6 Iceberg freeboard change

When icebergs are adrift, their motion is sufficiently large to mean that they are only sampled in 1-dimensional profiles along satellite altimeter ground tracks (Fig. 3) and only the largest tabular icebergs are sampled frequently enough to derive changes in their freeboard. We extract surface heights over the B30 iceberg when it is adrift (e.g. Fig. 5) using the position from the AIT database as an initial estimate of its location. However, because the AIT positions and timings are approximate and the iceberg has a significant extent, we investigate all CryoSat-2 ground tracks that pass within 1-degree latitude and 2-degrees longitude of the database position. We automatically extract measurements sampling the iceberg with the following steps: Track segments are truncated to exclude altimeter echoes from targets where the first or last freeboard height is more than 3 m, to exclude measurements from the nearby continent, and we also exclude tracks that do not contain freeboard measurement between 20 and 60 m, to ensure that they sample the iceberg. We consider all freeboard heights between the first and last echo falling in the range of 20 to 60 m as potential iceberg measurements (Tournadre et al., 2015). To avoid including adjacent icebergs or berg fragments, we exclude segments with more than 10 measurements of ocean or sea ice, identified as surface heights in the range -3 to +3 m, between potential iceberg measurements. We also remove crevasses and other rugged features using the same editing steps applied to determine the surface height prior to calving. As a final check, we calculate the distance of these remaining heights to the AIT database location, and discard measurements that are further away than half the iceberg length (28 km) to ensure we are tracking B30.

We apply two different techniques to calculate changes in the iceberg freeboard. For 12 tracks we are able to calculate precise changes in freeboard with spatial definition by making use of near-coincident satellite imagery to account for the rotation and translation of the iceberg relative to its initial position prior to calving (Jansen et al., 2007) and consider the estimated movement between the time of the nearest satellite image and altimeter acquisitions. At 94 other times, we compute the freeboard height change as the difference of mean freeboard from each new overpass relative to the initial mean surface height. While these observations are of poorer certainty, they provide denser temporal sampling and fill gaps between the colocated measurements. The first colocation method assigns both the initial heights and the new measurements to their closest 5 km grid cell and averages them to ensure that the same locations are compared. We account for the iceberg drift between the times of the satellite acquisitions, allowing a maximum separation of 72 hours (though most overpasses are separated by less than 24 hours). If the image is from a different date than the CryoSat track, we correct the distance travelled based on the daily iceberg locations from the AIT database. In any case, we account for the drift in our uncertainty estimate performing a Monte Carlo simulation with 1000 slightly differently collocated samples per track. These are normally distributed around our estimated translation and rotation with a standard deviation of 15° per day and a drift speed of 3 km per day (Scambos et al., 2008) scaled by the respective time separation. We then calculate the freeboard difference for each of the 1000 slightly differently colocated tracks and use the resulting standard deviation of freeboard change from these samples as the uncertainty of our colocation. This is combined with the standard deviation of the gridded CryoSat-2 freeboard data (of the new track and of the reference) to yield a conservative uncertainty estimate for the colocated tracks. The second method ignores the relative position and orientation of the iceberg at the time of the altimeter overpasses (Bouhier et al., 2018; Tournadre et al., 2015), and simply compares the mean freeboard along each new track to the mean surface height before calving. Although this method is easiest, since it does not rely on additional image data to locate the track, it cannot account for potential spatial variations in the iceberg freeboard. Because of this, we restrict the new overpasses to those including at least 20 measurements, as tracks sampling only the edges of an iceberg tend to be inaccurate. As uncertainty estimate we combine the standard deviation of each new overpass with the standard deviation of the initial height. As a first check to see if the mean freeboard from a single overpass can be compared to the mean initial height, we calculate the mean height for each of the 15 tracks over the pre-calved iceberg (Fig. 4a) and find a standard deviation of 2.8 m compared to the mean initial height of $49.0 \pm 4.6$ m.

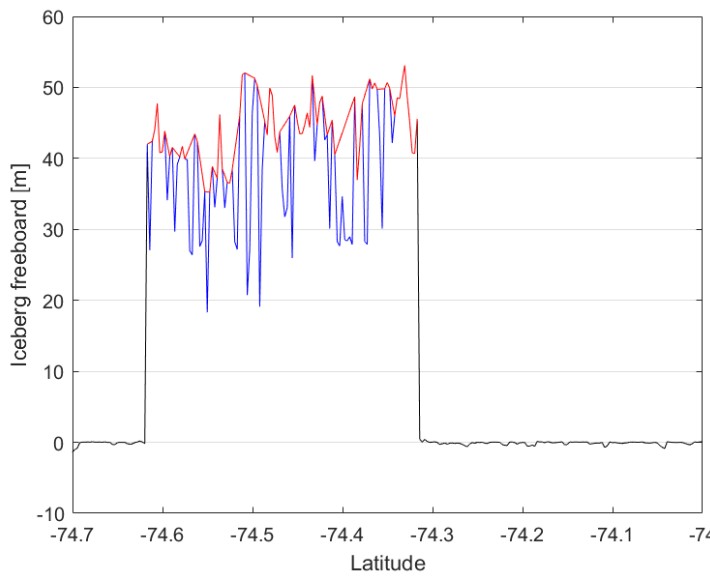

**Figure 5: Example of CryoSat-2 freeboard measurements along one track. The blue line shows which heights were identified as iceberg and the red line shows the remaining heights after filtering out crevasses.**

### 2.7 Iceberg thickness

We compute iceberg thickness $H$ (freeboard plus draft) from our estimates of iceberg freeboard heights $h_{fb}$ assuming hydrostatic equilibrium and that CryoSat-2 does not penetrate through the snow layer (Eq. 1; Moon et al., 2018). Besides these freeboard heights, iceberg thickness also depends on column-average densities of sea-water $\rho_w$, ice $\rho_i$, and snow $\rho_s$ as well as snow depth $h_s$. Including a snow layer in this equation is important, because the snow layer adds to the observed freeboard and disguises a part of the ice freeboard change. On the other hand the additional load of the snow layer pushes the iceberg downwards. Both effects are taken into consideration. We assume sea-water density to be 1024 kg m$^{-3}$ (Fichefet and Morales Maqueda, 1999) and set its uncertainty to 2 kg m$^{-3}$. Due to the long life cycle of the B30 iceberg of 6.5 years and the changing environmental conditions it experiences during this time, we allow the ice and snow densities to evolve with time. Snow depth is also time-varying, and estimates of this and of snow and ice density are introduced successively.

$$H = \frac{\rho_w}{\rho_w - \rho_i} h_{fb} - \frac{(\rho_w - \rho_s)}{\rho_w - \rho_i} h_s \qquad (1)$$

To estimate the thickness of the snow layer, we download hourly ERA5 Reanalysis snowfall, snowmelt, and snow evaporation data (Copernicus Climate Change Service, 2018), accumulate it daily and interpolate it in space and time to the iceberg's trajectory. Snowmelt and snow evaporation are subtracted from the snowfall to retrieve the additional snow accumulation since calving. However, this snow estimate does not account for snow being blown off the iceberg or onto

the iceberg from the continent (Fedotov et al., 1998; Leonard and Maksym, 2011). To convert snow water equivalent (SWE) to snow depth, we need to know snow density.

Snow density is time variable because snow compacts gradually during the iceberg's life time of several years as a function of snow depth $h_s$ [m], the mean air temperature $T$ [°C], and the mean wind speed $v$ [m $\cdot$ s$^{-1}$] (Eq. 2; International Organization for Standardization, 1998). We use hourly ERA5 Reanalysis 2 m air temperature data and calculate wind speed from the ERA5 Reanalysis 10 m eastwards and northwards wind components (Copernicus Climate Change Service, 2018). Both are interpolated to the iceberg's trajectory and averaged since the day of calving. Because snow density

depends on snow depth and snow depth depends on snow density, we calculate both iteratively starting with a snow density of 300 kg m$^{-3}$. We set the uncertainty in snow density to 50 kg m$^{-3}$ (Kurtz and Markus, 2012) and the uncertainty in snow depth to 20% (Kwok and Cunningham, 2008).

$$\rho_s = \left(90 + 130 \cdot \sqrt{h_s}\right) \cdot \left(1.5 + 0.17 \cdot \sqrt[3]{T}\right) \cdot \left(1 + 0.1 \cdot \sqrt{v}\right) \tag{2}$$

To calculate the iceberg's ice density profile we follow the approach by Tournadre et al. (2015), and determine two

parameters V and R to fit the surface density and the depths of the critical density levels (550 kg m$^{-3}$ and 830 kg m$^{-3}$) of the Thwaites Ice Shelf, from which it calved, as given in Ligtenberg et al. (2011; Eq. 3). $\rho_g$ is the density of pure glacial ice (915 kg m$^{-3}$). Since the mean ice density depends on ice thickness and ice thickness depends on the mean ice density, we iterate over both equations. We also account for ice density changes over the iceberg's life cycle by calculating new mean densities as the iceberg thins. This incrementally reduces the average ice density as the densest ice is melted at the

bottom. As ice density uncertainty we take 10 kg m$^{-3}$ (Dryak and Enderlin, 2020).

$$\rho_i = \frac{1}{H} \int_0^H \left(\rho_g - V \cdot e^{R \cdot z}\right) dz \tag{3}$$

## 3    Results and discussion

We first assess changes in the B30 iceberg area using boundaries mapped from satellite imagery, and we compare the observed trend to less accurate estimates derived from arc-lengths and semi-major axes. Next, we determine the change in

iceberg freeboard and we assess the impact of employing precise colocation using near-coincident satellite imagery. Iceberg thickness changes are then computed from freeboard changes using time-varying estimates of snow accumulation and snow and ice densities derived from atmospheric reanalyses. Finally, iceberg area and thickness changes are combined to derive the change in volume and mass.

### 3.1 Iceberg area change

When the B30 iceberg first calved in May 2012, it was $1500 \pm 60$ km$^2$. Over the following 6.5 years it lost $1075 \pm 66$ km$^2$ of its extent, which corresponds to a $72 \pm 11$ % reduction at an average rate of $149 \pm 5$ km$^2$ per year (Fig. 6). However, because deriving iceberg outlines requires a high degree of time-consuming manual interaction, we also evaluate the efficacy of two alternative methods based on measurements of their orthogonal (semi-major and semi-minor) axes by the NIC and on arc lengths recorded in satellite altimetry which are considerably less laborious. Although these approaches

also yield progressive reductions in area (Fig. 6), they exhibit significant positive (138 km$^2$, 14%) and negative (-426 km$^2$, 45%) biases, respectively, due to under-sampling of the iceberg geometry and the necessary approximation of a regular shape (ellipses and circles, respectively). While an ellipse overestimates the area compared to most shapes with the same axes, arc lengths yield an underestimate because corners are sampled more often than the major axis. One idea for improvement would be to use the maximum or to filter out tracks that only sample one corner, but the main problem

remains that a one-dimensional length measurement cannot be translated into a reasonable area estimate without knowing the iceberg shape, which changes over time. Nevertheless, both the orthogonal axes and arc-length approaches yield area estimates that are reasonably well correlated (r>0.90) with those determined from our manual delineation. Area trends are overestimated by 16% and underestimated by 48%, respectively. While manual delineation provides the most consistent and most accurate area estimate, tracking iceberg axes or arc lengths yields area and area change estimates that are within

48% and is considerably less time consuming.

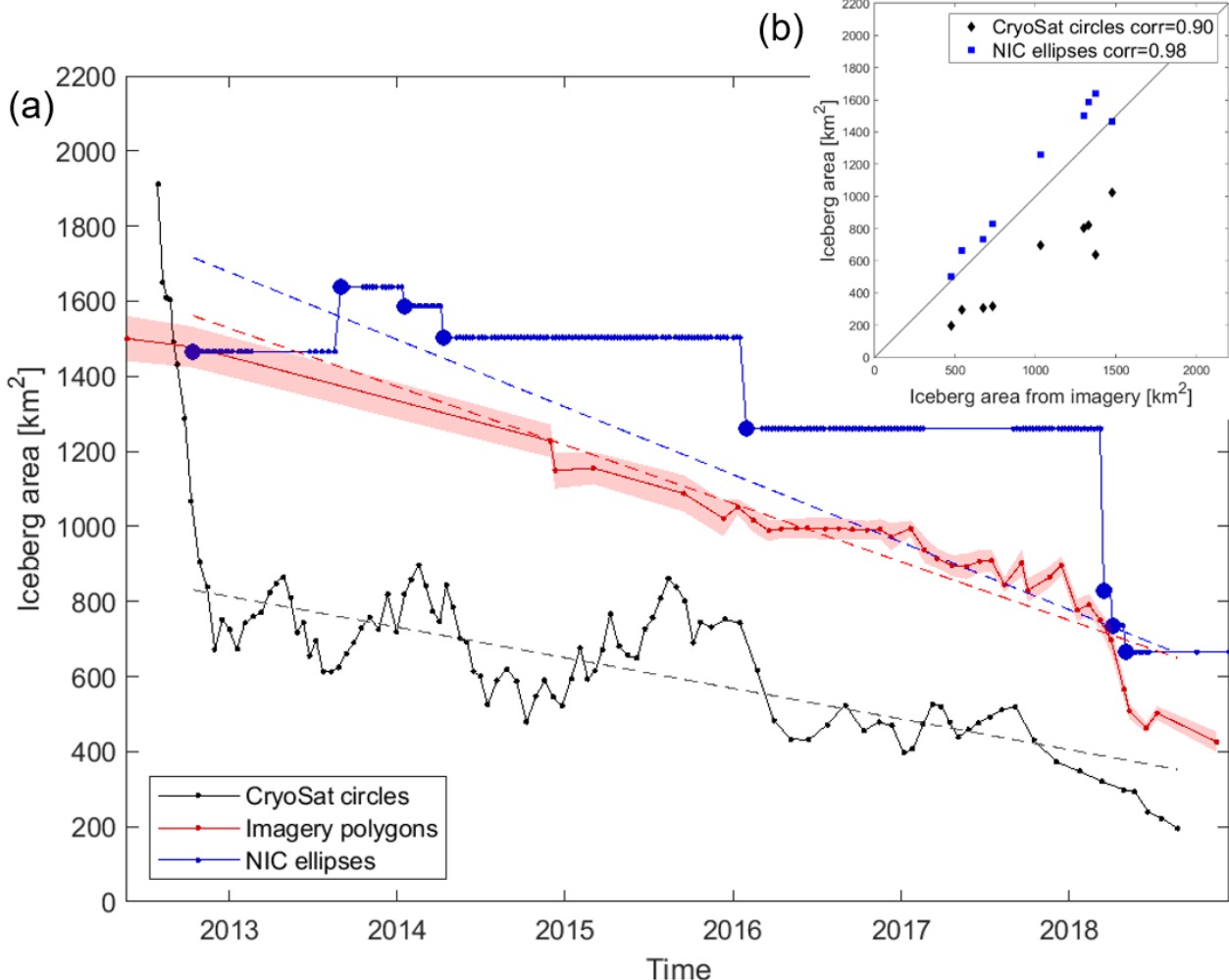

**Figure 6:** Area change of the B30 iceberg from polygons delineated in satellite imagery with their uncertainty (red) and approximations using orthogonal axes provided by the National Ice Center (NIC) assuming an elliptical shape (blue) or using the arc lengths of CryoSat-2 overflights assuming a circular shape (black) over time (a) and as scatter plot (b). To fit the NIC trend line in (a) we only use unique values of orthogonal axes length (thick blue dots). These also define the dates of comparison in (b).

The rate of iceberg area loss from B30 was approximately constant until 2018, after which time it started to lose larger sections more rapidly. Although its area has reduced steadily over time, it is less obvious which sections have been lost during individual calving events. However, by aligning the initial polygon to each subsequent image (Fig. 3) it is possible to identify when and where changes occur. The iceberg shape already appears altered on 30th November 2014, after bumping into the adjacent ice shelf which likely caused the first chunks to break off. B30 continued to lose smaller sections

along its edges over the next year – either through melting at the sides or smaller wastings – when it was drifting along the coastal current. In 2018, bigger sections are lost more rapidly, as the iceberg is drifting northwards in open water. Rift calving can occur at any time within an iceberg life cycle along pre-existing fractures (Scambos et al., 2008), while edge wasting is typically only observed when icebergs are travelling outside the sea ice pack. B30 was heavily crevassed prior to calving (e.g. visible in Fig. 3g and i), and so even the smaller wastings along its edges could reflect rift calving events rather than edge wastings. The 'footloose mechanism' (Wagner et al., 2014) can become a main driver of iceberg decay in warm waters, when wave erosion at the waterline forms a sub-surface foot, creating a buoyancy stress that can lead to calving. Although it is not possible to investigate the effects of wave erosion using satellite data, the effect could in principle have caused the larger break-ups that occurred in 2018.

## 3.2 Iceberg freeboard change

To assess the change in freeboard over the survey period, we compare differences between the new overpasses and the initial heights in space and time (Fig. 7). For the spatial analysis we chart the freeboard difference between each colocated overpass post-calving (Fig. 3) and the gridded initial height pre-calving (Fig. 4b) at the same relative iceberg position. This comparison shows that the change in freeboard height across the iceberg is relatively homogenous at each epoch (Fig. 7a-l). We then average these differences per CryoSat-2 track and chart the variation over time alongside the less accurate (but more abundant) estimates determined without colocation (Fig. 7m). Because the observations without colocation are relatively imprecise, we apply a 10-point moving mean to the data and we also fit a polynomial of $3^{rd}$ order (and starting at zero). Overall, the B30 iceberg freeboard has reduced by $9.2 \pm 2.2$ m during the 6.5 years since it calved.

To assess the importance of colocation, we compare freeboard changes calculated with and without this step (Fig. 7n). The estimates are well correlated (r=0.87) and the root mean square difference is 1.6 m, which is a measure of the improvement in certainty associated with colocation and equal to the difference in mean uncertainty of colocated tracks (4.7 m) versus tracks without colocation (6.3 m). Also, the temporal variation of freeboard changes computed from observations with and without colocation are in good overall agreement (Figure 7m), and we conclude that for this iceberg we can combine the two and make use of the entire set of CryoSat-2 measurements. This finding should hold for other tabular icebergs where the topographic variability is smaller than the observed thinning. The variability of freeboards computed within each 5 km grid cell and across different grid cells are also of the same order (3.3 m and 3.1 m, respectively), and this is likely to have reduced the impact of colocation uncertainties. For other icebergs with more heterogeneous freeboard across the iceberg that are less crevassed (i.e. with lower freeboard variabilities within the same grid cell), colocation might have a larger impact and more icebergs need to be studied to generalise these findings.

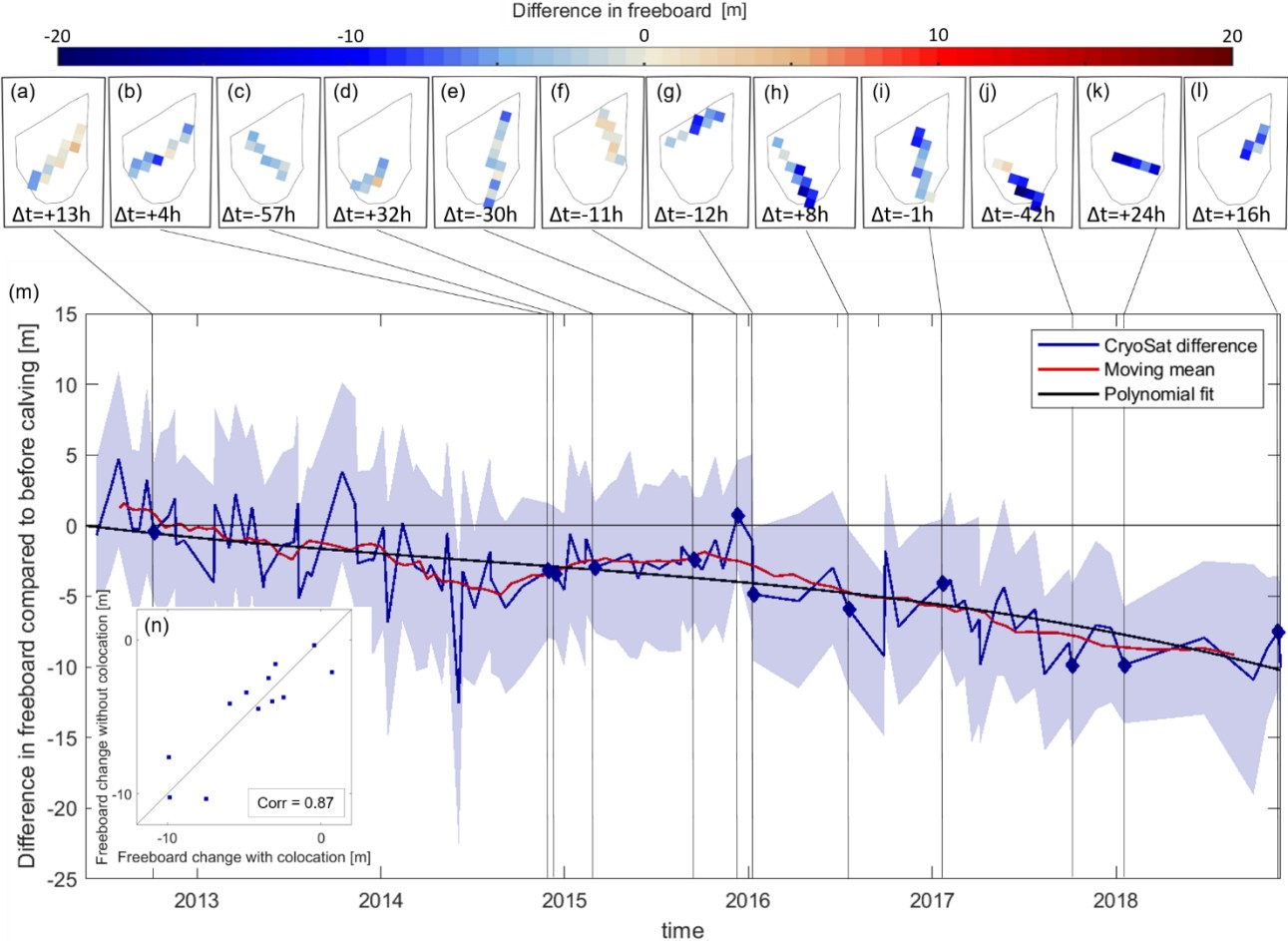

**Figure 7: Freeboard change of the B30 iceberg. a-l)** Freeboard difference in each grid cell sampled by colocated CryoSat-2 overpasses; the Δt values give the time difference between the CryoSat-2 overpass and the corresponding satellite image as an indication of the colocation uncertainty due to iceberg drift; Negative values indicate that the image was taken before the CryoSat overpass. **m)** mean difference of each new overpass along time. CryoSat-2 tracks that have been colocated are marked with a diamond, but all available CryoSat-2 overpasses have been used to calculate a moving mean and fit a polynomial; The shading shows the standard deviations. **n)** scatter plot of freeboard change from colocated CryoSat-2 tracks versus the same tracks used without colocation

### 3.3 Iceberg thickness change

We compute the iceberg thickness from our measurements of its freeboard (using the moving mean, red line in Fig. 7m) and by assuming that it is floating in hydrostatic equilibrium within the surrounding ocean with a surface snow layer.

Accounting for the snow layer is important because it affects the ice freeboard and the iceberg buoyancy, and we take both effects into consideration. Based on hourly snowfall, evaporation and snowmelt derived from ERA5 reanalyses (Copernicus Climate Change Service, 2018), we estimate that the iceberg accumulates 4.6 m of snow water equivalent during the 6.5 year survey period (Fig. 8). The rate of accumulation is quite linear. The iceberg thickness also depends on densities of the snow layer, the iceberg, and the sea-water and we allow the snow layer and iceberg densities to evolve over time due to the changing environmental conditions it experiences during its long lifecycle. The mean iceberg density reduces from an initial estimate of 864 kg m$^{-3}$ to a final value of 835 kg m$^{-3}$ as a consequence of basal ice melting (Fig. 8a). The mean change in height due to firn densification in West Antarctica has been estimated to be 2.79 cm per year on floating ice (Zwally et al., 2005); upscaling this rate gives a total of 18 cm after 6.5 years, which is significantly smaller than the observed freeboard loss of 9.2 m, so we don't apply it. The snow layer compacts over time due to its accumulation and warming, and we estimate that its average density rises from 252 to 616 kg m$^{-3}$ which yields a 7.2 m thick layer after 6.5 years (Fig. 8b). We also investigate the impact of surface thawing; although the iceberg surface does experience temperatures above freezing every summer and for a total of 218 degree hours (number of hours above zero degrees Celsius times the temperature above zero degrees Celsius) since calving (Fig. 8c), in situ observations (Scambos et al., 2008) suggest that this translates into only 8 to 16 cm of snow melting and this has negligible impact on the iceberg freeboard, so we discard this effect.

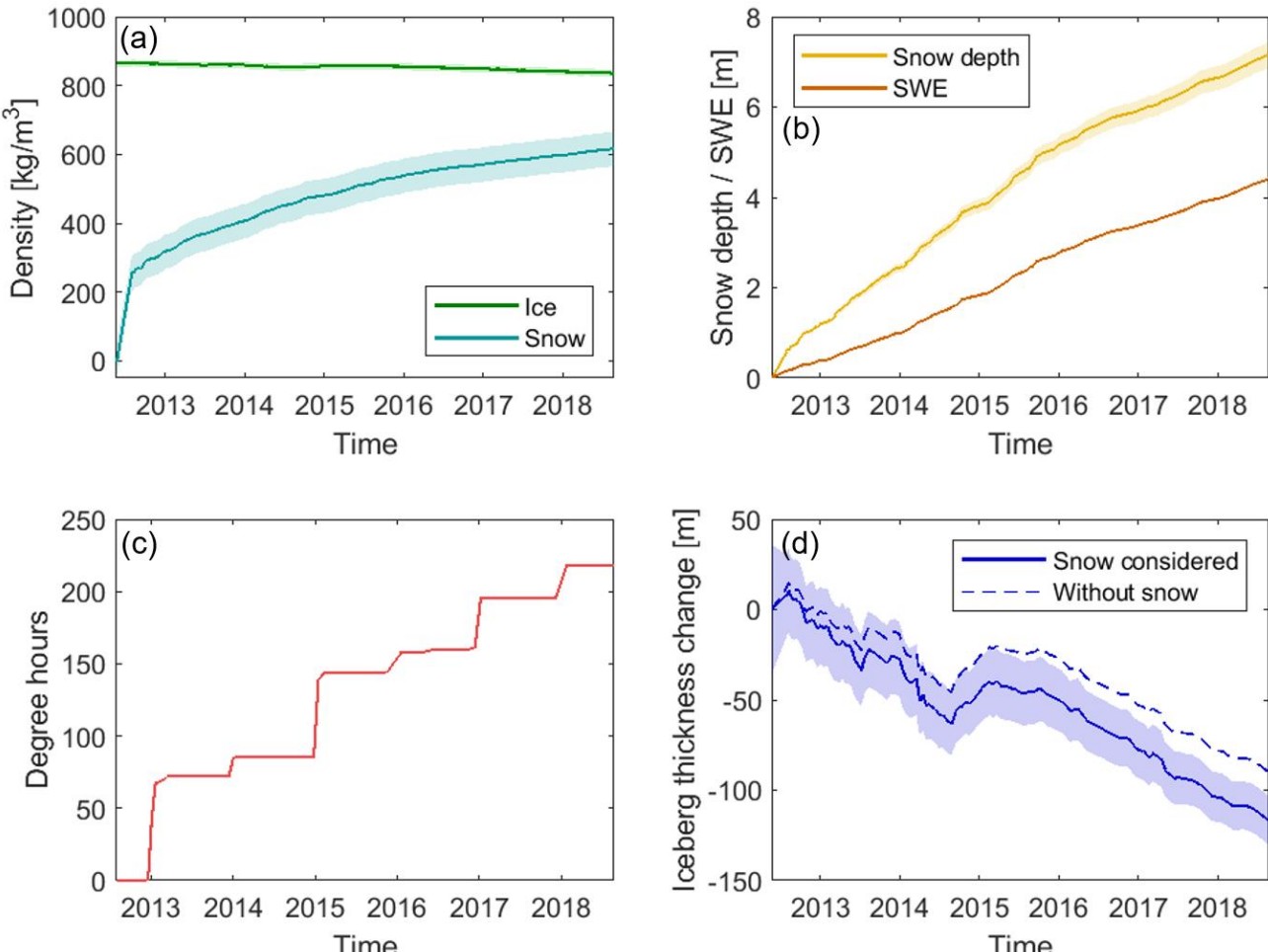

**Figure 8: Evolution of the B30 iceberg properties: a) Ice density and snow density, b) Snow water equivalent (SWE) and snow depth accumulation on the B30 iceberg, c) Degree hours that the B30 iceberg experienced, and d) Thickness change of the B30 iceberg with snow accumulation taken into consideration or without. Uncertainties are plotted as shaded areas.**

We estimate the initial iceberg thickness to be $315 \pm 36$ m, on average, reducing to $198 \pm 14$ m after 6.5 years. This amounts to $117 \pm 38$ m of thinning (Fig. 8d) at an average rate of $17.3 \pm 1.8$ m per year. Previous studies have recorded iceberg thinning rates of up to 10 m per year when drifting within the sea ice extent close to the coast (Han et al., 2019; Jansen et al., 2007; Li et al., 2018; Morgan and Budd, 1978; Scambos et al., 2008) and much higher rates in excess of 20 m per year when in warmer open water (Hamley and Budd, 1986; Jansen et al., 2007; Li et al., 2018; Morgan and Budd, 1978; Scambos et al., 2008; Tournadre et al., 2015). Jacka and Giles (2007) find dissolution rates of 11-18 m per year between 60 and

150° E based on shipborne observations over 15 years. Although all these studies were conducted for different regions of the Southern Ocean, our estimated average thinning rate is in line with the melt rates previously reported, given that the B30 iceberg has spent most of its lifetime close to the coast (Fig. 1). To assess the impact of including a snow layer in the
thickness calculation, we also compute thickness change assuming no snow has accumulated since calving (Fig. 8d); this scenario leads to an estimated 90 ± 39 m reduction in iceberg thickness, 23 % lower than the rate determined when the snow layer is included, which illustrates its importance. We expect the importance of including a snow layer to be highest in phases where the iceberg is melting slowly, as snow accumulation can disguise the thickness change in this instance. Based on the mostly linear snow accumulation, it will also be more important the longer the iceberg survives, as more snow
accumulates. Apart from the snow layer, iceberg density is also a significant factor in our thickness change calculation, and while we have attempted to model the evolutions of ice density, snow density, snow accumulation, and surface thawing, their uncertainties are difficult to quantify.

Besides the observed thinning, the iceberg also seems to slightly thicken between mid-2014 and early 2015. During this time B30 was very close to the coast (Fig. 3b-d). Therefore, a range of processes – both physical processes that impact the
actual thickness of the iceberg and processes that impact the freeboard measurement – could have caused this gain in thickness: First of all, iceberg thickness can increase through marine ice formation, when the iceberg is surrounded by very cold water. Little et al. (2008) found that freezing beneath ice-shelves is concentrated along their western side and B30 was indeed located at the western side of Getz Ice Shelf at this time (Fig. 1, 3b, c). Iceberg thickness can also grow through snow accumulation on the surface, which we account for, but only based on reanalysis data and there might be additional
local snowfall or snow accumulation through strong katabatic winds from the near-by continent (Fedotov et al., 1998). Furthermore, external forcing from collisions with the adjacent ice-shelf might have led to a deformation (MacAyeal et al., 2008) and hence a compression in some parts. All of these processes can cause a physical increase in iceberg thickness. Apart from that, a short (partial) grounding could lead to higher measured iceberg freeboards (Li et al., 2018). Also surface melting could shift the scattering horizon of CryoSat-2 (Otosaka et al., 2020) and therefore appear like a freeboard increase.
Indeed we observe a steep increase in degree hours around the turn of the year 2015. What caused the signal in this instance is hard to disentangle. Most probably, it was a combination of several of the mentioned effects.

## 3.4 Iceberg volume and mass change

Having calculated changes in the B30 iceberg thickness associated with snowfall and basal melting and changes in area due to fragmentation, we combine both to determine the overall change in volume (Fig. 9). To do this, we multiply each
thickness estimate with the imagery-based area estimates interpolated to the times of the CryoSat-2 overpasses. Unlike small icebergs, which can take on various shapes (Enderlin and Hamilton, 2014; Sulak et al., 2017), large tabular icebergs

inherit their shape from their parent ice shelf and therefore have rather homogenous thickness and near vertical walls (American Meteorological Society, 2012). Deviations from vertical may occur in both directions and we therefore expect them to approximately even out (Orheim, 1987). The larger the length to thickness ratio is, the smaller the impact of tilted side walls on the resulting volume. For the B30 iceberg with an initial length to thickness ratio of 187:1, we therefore conclude that our assumption of vertical walls has negligible impact on the volume. The proportion of the total volume changes associated with melting and fragmentation are calculated by keeping area and thickness constant (and equal to their average), respectively. To compute changes in mass, we multiply the volume change due to fragmentation by the column-average iceberg density at each point in time, because this ice is lost at the sides. In contrast, we multiply the volume change due to basal melting by the density of pure ice (915 kg m$^{-3}$), since this ice is lost at the bottom where ice density is highest. The total mass change is the sum of both components. Uncertainties are calculated by propagating the uncertainties of thickness change, area change, and ice density.

The initial volume of B30 at the time of its calving was $472 \pm 57$ km$^3$ and after 6.5 years it has lost $378 \pm 57$ km$^3$ of ice, corresponding to a $80 \pm 16$ % reduction. Fragmentation accounts for two thirds ($69 \pm 14$ %) of the total volume loss and basal melting is responsible for the remainder ($31 \pm 11$ %). Volume changes due to fragmentation become the dominant source of ice loss towards the end of our survey, consistent with previous findings (Bouhier et al., 2018). This is because the main drivers of fragmentation are surface melting, which can lead to a rapid disintegration (Scambos et al., 2008) and wave erosion or wave stress (Wagner et al., 2014). Both increase the further North (i.e. surrounded by open ocean and warmer air temperatures) the iceberg gets. The two icebergs studied by Bouhier et al., (2018) also show similar fractions of ice loss due to fragmentation (60% for the B17a iceberg and 75% for the C19a iceberg). In terms of mass, the iceberg has lost $325 \pm 44$ Gt of ice in total at an average rate of $46 \pm 4$ Gt per year. The loss due to basal melting ($106 \pm 35$ Gt) can be used as a lower estimate of the freshwater flux from B30. Some of the mass lost due to changes in area - in particular melting at the sides and smaller edge wastings, which will probably melt locally, add to the freshwater flux, but bigger calving events create smaller icebergs, which can survive and travel on their own (Bigg et al., 1997; England et al., 2020; Martin and Adcroft, 2010). To calculate the total freshwater flux, the melting of all fragments has to be considered (Tournadre et al., 2012, 2016).

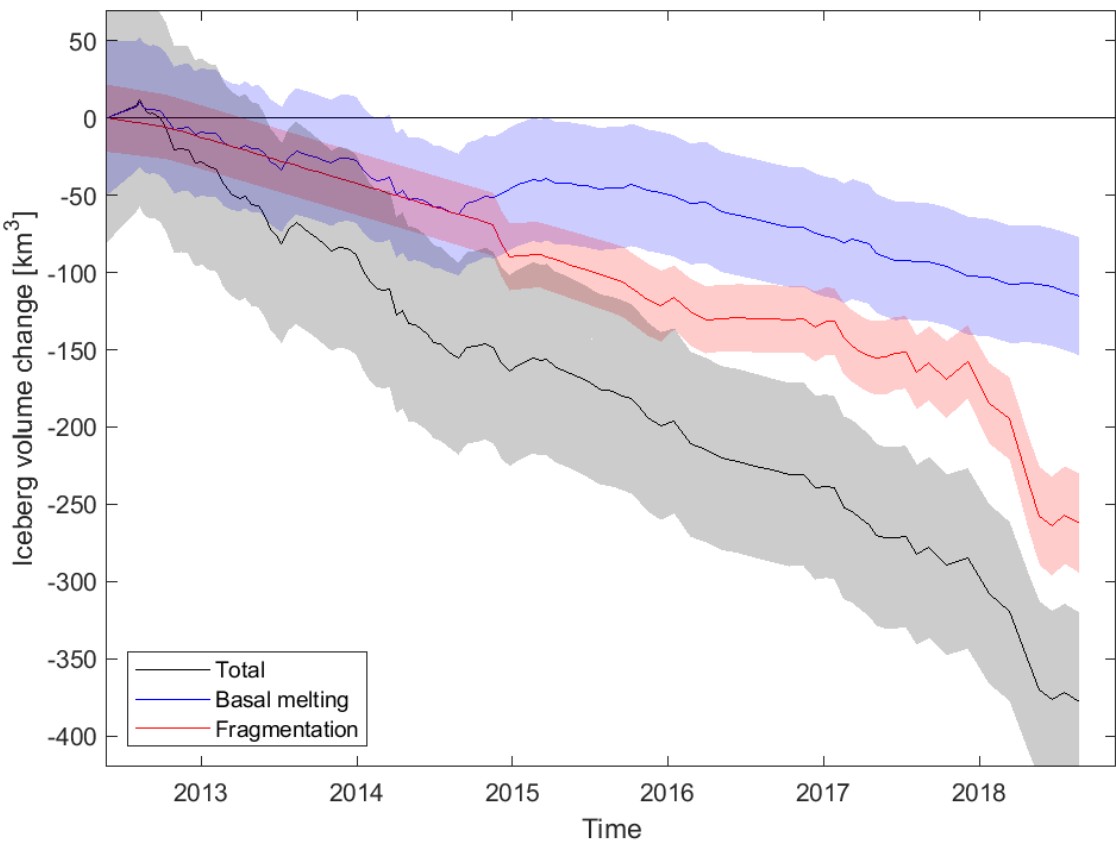

**Figure 9: Volume change of the B30 iceberg divided into loss due to basal melting (thickness change, blue) and due to fragmentation (area change, red), as well as total volume loss (black).**

## 4 Conclusions

In this study we have derived changes in the area, freeboard, thickness, and volume of the tabular B30 iceberg using a combination of satellite altimetry and satellite imagery. During the 6.5 years after the iceberg calved in May 2012, its area reduced from $1500 \pm 60$ km$^2$ to $426 \pm 27$ km$^2$ at an average rate of $149 \pm 5$ km$^2$ per year. The iceberg freeboard lowered by $9.2 \pm 2.2$ m over the same period. Using estimates of the snow accumulation and changes in snow and ice density, we estimate that the iceberg thinned by $117 \pm 38$ m at a mean rate of $17.3 \pm 1.8$ m per year. Altogether, the iceberg lost $378 \pm 57$ km$^3$ of ice, and this equates to an estimated $325 \pm 44$ Gt reduction in mass.

We investigated the capability of automated approaches to approximate iceberg area and area change by comparing them to manually-derived estimates. Although the most reliable method of charting iceberg area change is through manual delineation in satellite imagery, we show that less time-consuming estimates derived from measurements of the iceberg's orthogonal axes or arc-lengths are also able to capture the area and area change over time, albeit with poorer certainty. Orthogonal axes lead to estimates of area and area trends that are 14 % and 16 % higher, respectively, and arc-lengths lead to estimates of area and area trends that are 45% and 48% lower, due to the necessary approximate of the iceberg shape.

We also presented a new thorough methodology to investigate iceberg freeboard and thickness change, using a densely sampled time series of consistently processed Level 1 CryoSat data and assessed the importance of colocation. Using a subset of 12 instances with colocation, we find that omitting this step leads to a small deterioration in the certainty of detected freeboard change for the B30 iceberg, but the densely sampled time series is in good agreement with the colocated tracks. We expect this finding also holds for other large tabular Antarctic icebergs with uniform topography, when the observed freeboard change exceeds the topography and when enough tracks are averaged. In this case, it suggests that the procedure for tracking changes in iceberg thickness could be automated, given reliable estimates of their position (Budge and Long, 2018).

Finally, we developed a methodology to account for snowfall and variations in snow and ice density due to changing environmental conditions that large icebergs experience during their multi-annual drift. We found that the impact of snowfall on the retrieval of iceberg thickness increases over time, and after 6.5 years we estimate that 7.2 metres of snow have accumulated, which leads to a 27 m adjustment to the iceberg thickness change. Iceberg thickness change is also strongly dependent on the ice density profile which we derive from the depths of critical density levels (Ligtenberg et al., 2011), and so in situ observations would help to assess the reliability of this relationship. Likewise, direct measurements of the near-surface firn will help to assess the reliability of our reanalyses-based estimate of snow loading.

More icebergs - including the fragments lost from B30 - need to be studied to generalise the results we have and to constrain both the fresh water flux, which influences water circulation (Grosfeld et al., 2001; Jenkins, 1999) and promotes sea ice formation (Bintanja et al., 2015; Merino et al., 2016), and input of terrigenous nutrients such as glacial iron into the Southern Ocean, which fosters primary production (Biddle et al., 2015; Duprat et al., 2016; Helly et al., 2011). Finally, studying icebergs as they drift through warmer water may give unique insights into the response of glacial ice to environmental conditions which may become commonplace at the ice shelf front in the future (Scambos et al., 2008; Shepherd et al., 2019).

**Code availability**

The code (mostly written in matlab) is available from the authors upon reasonable request.

**Data availability**

All data used in this study is freely available: The iceberg trajectory data is available from https://www.scp.byu.edu/data/iceberg/, CryoSat-2 data is available from https://science-pds.cryosat.esa.int/, Sentinel-1 data from https://scihub.copernicus.eu/dhus/, MODIS data from https://ladsweb.modaps.eosdis.nasa.gov/search/order/1/MOD02HKM--61, and the ERA-5 reanalysis data from https://cds.climate.copernicus.eu/cdsapp#!/dataset/reanalysis-era5-single-levels.

**Author contributions**

ABF and AS designed the study, AR processed the CryoSat elevations, ABF computed freeboard, area, and volume change and prepared the figures, AS supervised the work. All authors contributed to the writing.

**Competing interests**

The authors declare that they have no conflict of interest.

**Acknowledgements**

This work was supported by Barry Slavin and the Centre for Polar Observation and Modelling. The Antarctic Mapping Toolbox (Greene et al., 2017) was used to convert geographic coordinates to polar stereographic and vice versa and to calculate distances. We thank the anonymous reviewer and Jessica Scheick for their detailed reviews and Xuying Liu for a short comment, which all helped to improve the manuscript.

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
