# Peer review of "Tracking Changes in the Area, Thickness, and Volume of the Thwaites Tabular Iceberg ‘B30’ using Satellite Altimetry and Imagery"

_The Cryosphere, 2021_

## Referee Comment (RC1)

Journal: TC

Title: Changes in the Area, Thickness, and Volume of the Thwaites "B30" Iceberg Observed by Satellite Altimetry and Imagery

Author(s): Anne Braakmann-Folgmann et al.

MS No.: tc-2021-13

MS type: Research article

**General comments**

The article *"Changes in the Area, Thickness, and Volume of the Thwaites "B30" Iceberg Observed by Satellite Altimetry and Imagery"* presents area, thickness, and volume variations of the iceberg B30 along its lifespan. To get such estimates, the authors rely on different remote sensing products – altimetry from CryoSat-2 and optical and radar imagery from MODIS and Sentinel 1, respectively. The article is, in general, very well written and thorough, and brings new data to iceberg science, a growing field of research. Such data are necessary to better parameterize numerical models that intend to study the role of icebergs in freshwater distribution in the ocean. My most significant comments relate to the author's definition of geolocation, and if this geolocation is indeed worth the extra amount of work given its questionable uncertainties. I conclude that, with a clearer explanation of the methodology and a few corrections, this work will be ready to be published – and will be a welcomed contribution for The Cryosphere.

**Specific comments**

Line 13:    What do you mean by "*different modes*"?

Line 15:    "*compare this time series to precisely located tracks using the satellite imagery*"
This sentence and the further use of "***geolocation***" in the paper confuses me. I'm not a person that works with remote sensing, but as far as my understanding goes (please correct me if I'm wrong), altimetry data (at least in its final product) should have latitude-longitude information associated with each data point. And, to my mind, geolocation means using some land feature with known coordinates in a non-referenced image to infer the lat-lon points in that image. That wouldn't seem to be a problem with altimetry data. From what I have understood from the paper, by "geolocation" you actually mean being able to tell the orientation of the iceberg, so the points over the iceberg are compared with themselves even when the iceberg has rotated.

E.g.: Say you have points **n** and **m** assigned before calving ($t_0$); after $\Delta t$ amount of time drifting, CryoSat-2 measures the height indicated by the yellow "pixel". If you don't know the iceberg's orientation, you could assume the height taken refers to point n (as if the iceberg has not rotated, transparent image in the background. But if you to know the new orientation (by "geolocating" the iceberg using imagery), you see that the height measure actually refers to point m.

[Figure]

[Figure]

Did I understand this correctly? Either way, I think this needs to be clarified on the text. I would say something like "determine the iceberg's orientation" instead of "geolocate" or, in case you really mean geolocate, to explain that altimetry data is not referenced to lat-lon positions.

Line 26: Very nice overview paragraph!

Line 128: *"(ii) using measurements of the semi-major and semi-minor axes provided by the NIC **and assuming an elliptical shape** and (iii) using measurements of their arc lengths recorded in satellite altimetry **and assuming a circular shape**."* Just to be bluntly clear.

Line 176: "*to align all images to a common orientation (Fig. 3)*". From my point of view, the common orientation can be seen in Figure 7a-l, not in Fig.3. Unless you are talking about the altimeter track orientation – if so, please specify.

Figure 3: There is a **Okt** in figure (a) and a **Dez** in figure (c). I'd also indicate which sensor each image is from, which might not be obvious to a person that does not work with satellite data.

Line 237: How exactly are you accounting for the iceberg drift? Have you calculated the distance the iceberg has travelled between the CryoSat and MODIS/Sentinel measurements\*? And what does this imply? I assume you look at different locations between altimetry and imagery to find the iceberg (which has drifted between measurements). Or do you take the height measurements from the same points the iceberg is occupying in the imagery, assuming they didn't move but assigning an uncertainty to the drift?

\*In model simulations, the icebergs' average speed is 0.14 m/s, which means it would have moved around 500 m in 1h.

Then, you assume that the iceberg has kept the same orientation but account for an error of 15°/day. How do you combine all those uncertainties to get to a final error in the freeboard estimate? Could you include an equation or give more detailed information on the supplementary material?

Although the methodology is, in general, very well described, I'm not sure about this part. And that makes me wonder if using the imagery to infer spatial height variability (assigning

the altimetry data to specific points on the iceberg) is worth the trouble. I see that the final variability is reduced when this procedure is done (compared to the averaged freeboard estimate) on line 340, but as you mention, this is probably the case because the iceberg has a "homogeneous thickness", i.e., if you mistake one grid cell for another it is not a big deal. And, if the thickness is homogeneous, then the averaged height along the track (without "geolocation") should be good enough – which indeed is the case, since you mention that the different methodologies' results are in good agreement. So, maybe this "geolocation" is more useful when one is dealing with a non-tabular iceberg, where the thickness from point to point varies a lot and doing simply a freeboard average along the iceberg would lead to the significant loss of spatial information. However, in this case, the uncertainties brought by the time difference between the altimetry and imagery would be much more damaging to final results, since mistaking one point of the iceberg for another would imply much larger errors.

Line 253:   I assume by thickness you mean draft + freeboard?

Line 267:   Define SWE.

Figure 7:   The caption needs to be updated with the subplots' correct letters.
"*m) mean difference of each new overpass along time.*"
"*a-l) freeboard difference in each grid cell*"

Also, in a-l, the $\Delta t$ is indeed always positive? i.e., is the satellite image taken always after the CryoSat overpass? If not, you could differentiate them with a minus sign for images taken before the CryoSat overpass.

Figure 8:   (a), (b), (c), and (d) labels missing from plots

Line 376:   I'd actually move the reference to Figure 8d to the next sentence:
"*This amounts to 117 ± 38 m of thinning (Fig 8d)*"

Line 391:   Fig 8d actually shows the thickness **differences**

Line 400:   "*volume changes due to fragmentation become the dominant source of ice loss towards the end of our survey*". Isn't that funny, though? I'd imagine it is much easier to break a large piece of ice than a small one. And it even somewhat contradicts what was said in line 57: "breakage dominates over melting for large icebergs". Could you offer an explanation, then, why fragmentation becomes more important at the end of this iceberg's life?

Line 406:   A relevant reference here is Martin and Adcroft (2010) when talking about bergy bits.

**A note about the colormaps used in the figures:** although I do enjoy rainbow colormaps, it is good to think about inclusion – namely, colorblind readers. Even for me, the colormap in Figure 1 and 2 could get confusing: it starts with red and finishes with… red. Of course, we infer which color corresponds to which year just by following the progression of the iceberg, but that should be clear from the color scheme as well. The colormap in Figure 4c and 4d is a good one: starts with bright colors, finishes with dark ones. When in doubt, there are resources such as:

where you can upload you figure and see how it looks like for someone with color vision deficiency. Also, I prefer discrete color bars rather than continuous ones, so you can really see what values are attributed to each color. I don't expect you to change all your figures now, but I wanted to throw the idea out there to keep in mind for next publications.

**Technical corrections**

Line 27: Be mindful of commas. I found them missing from some places, but there could be more around. Here: "*At any time**,** (…)*".

Line 34: Check if there is a space between "*production*" and the following parenthesis.

Line 56: "*iceberg melting**,** to first order**,** (…)*" or rearrange the sentence such as "*found that iceberg melting is proportional to water temperature to first order (…)*".

Line 71: "*also include**d***" – just to be consistent with "*studied*" in line 70.

Line 73: "*employ**ed** altimetry measurements*" – same as above.

Line 125: "*The initial **area***" (also, could you provide the length of the iceberg?)

Line 74: "*Bouhier et al. (2018) analyse**d***" – same as above; plus, that's a long sentence. I'd just split it into 3 shorter ones, one for each citation.

Line 75: "*(…) geolocation**.** Li et al. (2018) calculate**d***"

Line 76: "*(…) overpasses**.** Han et al. (2019)*"

Line 78: "*When thickness and area changes are combined**,** it is possible (…)*"

Line 174: "*CryoSat-2 overflights**.***" – the end of this sentence doesn't read well, so you can just remove it.

Line 287: Maybe "less accurate" would sound better than "*more approximate*"

Figure 6: On the caption: "*over time (a) and as scatter plot **(b)**.*"

Line 315: "*larger sections  more rapidly.*"

Line 320: "*In 2018**,** (…)*"

Line 341: Do you mean Figure 7m instead of 5a?

Line 393: "*To compute changes in mass**,** we (…)*"

Line 412: "*thickness**,** and volume (…)*"

---

## Author Response (AR1)

*Thank you to the editor and the two reviewers for their positive and detailed reviews. Thank you also to Xuying Liu for her short comment and interest in the paper. We counted 133 individual comments in total. We have revised our manuscript to address 114 and we have explained misunderstandings to address the remaining 19. The main changes are:*

- *We rephrased "geolocation" to "colocation", as reviewer 1 correctly pointed out that the tracks are geolocated with respect to the earth, but we mean a colocation with the initial iceberg outline prior to calving*
- *We explain our choice to study the B30 iceberg at the end of the introduction*
- *We changed the colourmap of Figure 1 and 2*
- *We added 403 extra words to the methodology, including 12 additional references (6 new papers)*
- *We added 629 extra words to the discussion, including a paragraph on possible causes for iceberg thickness increase and 18 additional references (11 new papers)*
- *We added code and data availability statements*

*Please also see our responses to the specific comments and technical corrections in* blue *and the tracked changes attached at the end of this document. Line numbers mentioned in our responses refer to the original (Discussions paper) document. We believe that these changes have substantially improved the manuscript. Thank you again to the editor, both of the reviewers and Xuying Liu for their comments.*

**Initial comments by the editor (Johannes Fürst), 22 February 2021:**

*Responses and changes that were incorporated before the review are* light blue*, changes that were made with respect to the Discussions paper during review are marked in* blue*.*

- You present a rather concise comparison between the co-location technique and the non-geo-located reference detection. This information is rather utile in assessing the different techniques. Yet for the modelling part of your results, uncertainties remain vague and are not presented very consistently.

We found it quite hard to quantify the uncertainties especially in the freeboard to thickness conversion and couldn't find any studies that ever looked into this for icebergs. The closest studies are on sea ice, but not all components are transferable because e.g. the ice stems from glaciers and therefore has a different density than sea ice. Also, the snow on icebergs can get much thicker, because they survive and travel for much longer, but also because the snow load on an iceberg never gets flooded – unlike sea ice. We therefore put some effort into finding reasonable estimates, but do acknowledge that they are, at this point, more like a best guess. We also mention and would like to emphasize the need for in situ observations of iceberg density and the evolution of the snow layer on an iceberg, which are obviously hard to acquire, but could shed more light on this. In the paper we added additional references for the uncertainties used. In terms of the presentation we made the uncertainty shading more obvious and they should therefore appear more consistent now (see list of issues).

- Concerning the freeboard estimate, I am puzzled how you infer the volume changes. For this you either need 2D maps of the freeboard height, which you do not have. I therefore suspect that you simply average the freeboard point information along the CryoSat tracks (subsampled to the 5-km resolution). Please clarify in your description or make it more prominent.

True, we indeed average the freeboard height of each CryoSat overpass and assume the iceberg is melting equally across space – which seems a reasonable assumption looking at the maps of freeboard change from geolocated overpasses (Figure 7a-l). This is how we generate the time series of freeboard change and thickness change and the latter is then used to derive volume change. For more details, please see our response to the list of issues (L228).

- Possibilities/suitability for an automation of the presented techniques remain unaddressed. I am mostly intrigued by how we can transfer this monitoring to other icebergs. Which parts of your processing chain are automatable. You might therefore consider picking up this concern in your discussion chapter or add it to the outlook.

Indeed one main goal of this paper was to investigate which parts of the processing chain are automatable and to which extend the automation degrades the accuracy compared to more time-consuming manual alternatives. We are for example discussing this for area change in section 3.1, where we find that manual delineations are still much more accurate. For freeboard and thickness change we compare tracks without geolocation – i.e. without manual interaction – to geolocated tracks and find that they agree well, if enough tracks are available and if the iceberg's topography is of the same order of magnitude as the variation within each grid cell. We added more discussion on this in the paper. In the outlook we also mention that the transferability to other icebergs should be investigated in the future.

- You blended the 'Results & Discussion' section which I like but in my view the discussion part falls rather short. I therefor suggest disentanglement and a focused discussion of the results in view of other work.

We added more discussion to the manuscript including
- a discussion of the impact of colocation on different icebergs
- more extensive comparisons to the melt rates found by previous studies
- further discussion on the impact of snow accumulation on our thickness calculation
- a whole new paragraph on possible causes of iceberg thickness increase
- a discussion on the impact of iceberg shape on the iceberg volume calculation for small icebergs versus very large ones
- further discussion on the drivers of fragmentation.

We also added 18 additional references in the Results and Discussions section to clarify the limitations, differences and advances with regard to previous studies.

On the suggestion that we should divide our results and discussion, we prefer not to do this as our findings are naturally divided into four topics – iceberg area, freeboard, thickness, and volume change – and to disaggregate the results and discussion is disruptive to the flow of our text. It is not uncommon to aggregate results and discussion for this reason.

- Section 2.5 could be better split into a section on the initial freeboard map and the consecutive evolution.

Done.

- some minor editorial comments on the figures see below

Done.

LIST OF ISSUES to be addressed *(line refer to a previous version):*
- L45-48 What impact does a better representation of iceberg trajectories and volume changes have on climate modelling. Briefly specify.

We made this sentence more specific.

- L83-86 This passage sounds like you give an overview of the article. You could add at the end a brief overview of the manuscript structure. 1-2 sentences

We added a few sentences on this.

- L143 relative difference

Fine.

- L170 What do you mean by p,q? It becomes clearer afterwards. Think about reformulating.
- L171 From you description, I was unsure how the angle and reference points (p,q) are determined. My feeling is that you use a manual fitting procedure for the 12 images. Please explain
- Eq1. Consider removing this equation from the text. It describes a simple rotation of a coordinate system. Your description might well suffice.

We removed the equation and all explanations of the variables in the text before (e.g. p,q). You are right: It's a manual fitting procedure, where one point is clicked manually and the angle as second variable is adjusted until the initial polygon is well aligned with the image. We modified the description to make this clearer.

- L192 Somewhere in this section on 'outlier deletion', please state the primary aim on crevasse measurement removal. This motivation is only mentioned later but repeated often hereafter.

Done.

- L201-202 Give a reference to these extra in-situ freeboard height measurements. Did you acquire them yourself?

We rephrased this part to make it clearer that it's not in situ observations, but simply the first CryoSat overpass after calving. Because the iceberg is definitely afloat then, we can compare this overpass to the measurements when the iceberg was still attached and verify that our initial heights are comparable to measurements acquired over a free floating iceberg.

- L228 Concerning the co-location method, I got lost. As I understand it, you have a single CryoSat track as well as satellite image telling you about the orientation of the iceberg. I do not quite understand how you get a 2D freeboard height from that. Without geo-location, I have the same issue. Do you simply average the computed measurements of freeboard height no matter how the transect orientation to get a mean elevation change value. I might have missed this information. Please clarify.

You are right, we don't get a full 2D freeboard map from one overpass – just a 1D transect (see Figure 3). For the colocated overpasses the 1D transect is transformed to the initial reference system and gridded on the same grid as the initial heights. We then compare the new freeboard in each grid cell that was covered by the new overpass to the initial freeboard in the same grid cell (see Figure 7a-l). For the time series (Figure 7m), these difference maps are averaged per overpass (Lines 332-

334). Without colocation, we also simply average the new freeboard height per overpass and difference it with the mean initial freeboard (of the whole iceberg, because we don't know the exact location, lines 233-234).

- L251 Here you could check the World Ocean Circulation Experiment Southern Ocean Atlas to verify the density assumption.

1024 kg/m^3 is very commonly used as sea water density in studies on sea ice in the Southern Ocean (e.g. Fichefet and Morales Maqueda, 1999; Kacimi and Kwok, 2020; Zwally et al., 2008). As this value varies the least and has very little impact on freeboard to thickness conversion, we didn't put too much effort into verifying this value and keep it static. Including a sea water density uncertainty of 2 kg/m^3 is also more for the sake of completeness, and only contributes 2 % to the uncertainty budget of iceberg thickness.

- L324-326 A sentence on time difference is already in caption Fig.7. Consider removing it here.

Agreed.

- L384-391 The description of the partitioning into the mass loss due to fragmentation and due to basal melting remains a bit vague. Please clarify. You also do not discuss how these two categories relate to rift calving, edge wasting and rapid disintegration. These terms you introduced in the beginning. It might be worth to include the portioning you present here in the Methods section.

Concerning mass loss: We have fitted two parameters V and R to describe the iceberg's density profile as described in 2.6, equation 4. In case of fragmentation, the iceberg loses pieces of ice at the sides and therefore their density is the mean iceberg density at this point in time. In case of basal melting, the ice, however, is lost from the iceberg's base, where ice density – according to the ice density profile – is highest and equals 915 kg m-3. We clarified this in the text, too.

In general, fragmentation includes rift calving, edge wasting and rapid disintegration, since all of these processes are related to area change. Most of this discussion part can therefore be found in the section on area change (3.1) and we briefly also mention it at the end of the volume change section.

Fig.2a Scalebar and north arrow for orientation.

Done.

Fig.6b This inset is rather small and the labels are hard to read. Please consider producing a larger panel. You could increase it toward the upper right corner overlaying the frame of panel a.

This is a good idea. We modified the figure.

Fig.6 I wonder if the last sentence in the caption refers to panel (a). If so, please rearrange the caption accordingly.

We updated the figure caption.

Fig. 8 The ice thickness change in panel b has an uncertainty range which is not described in the caption. Moreover, I wonder about why there are no uncertainty estimates in panel a-b. You took

quite some effort to consistently indicate the input uncertainties that you assume. Please add or justify.

We have plotted uncertainties for the ice density, snow density and snow depth, too, but as they are smaller compared to the range of the y-axes, they weren't easily visible. We therefore reduced the transparency for all plots that have uncertainties plotted with them. Furthermore, we now also mention the uncertainties in the figure caption.

**Anonymous reviewer #1, 8 April 2021:**

**General comments**

The article *"Changes in the Area, Thickness, and Volume of the Thwaites "B30" Iceberg Observed by Satellite Altimetry and Imagery"* presents area, thickness, and volume variations of the iceberg B30 along its lifespan. To get such estimates, the authors rely on different remote sensing products – altimetry from CryoSat-2 and optical and radar imagery from MODIS and Sentinel 1, respectively. The article is, in general, very well written and thorough, and brings new data to iceberg science, a growing field of research. Such data are necessary to better parameterize numerical models that intend to study the role of icebergs in freshwater distribution in the ocean. My most significant comments relate to the author's definition of geolocation, and if this geolocation is indeed worth the extra amount of work given its questionable uncertainties. I conclude that, with a clearer explanation of the methodology and a few corrections, this work will be ready to be published – and will be a welcomed contribution for The Cryosphere.

Thank you for your positive review and your insightful comments on our manuscript – we have addressed each of your specific comments below and think that these changes truly improved the manuscript.

**Specific comments**

Line 13: What do you mean by *"different modes"?*

CryoSat-2 has three acquisition modes: Low Resolution Mode (LRM), Synthetic Aperture Radar (SAR) and Synthetic Aperture Radar Interferometric (SARIn). We clarified the sentence.

Line 15: *"compare this time series to precisely located tracks using the satellite imagery"* This sentence and the further use of *"**geolocation"*** in the paper confuses me. I'm not a person that works with remote sensing, but as far as my understanding goes (please correct me if I'm wrong), altimetry data (at least in its final product) should have latitude-longitude information associated with each data point. And, to my mind, geolocation means using some land feature with known coordinates in a non-referenced image to infer the lat-lon points in that image. That wouldn't seem to be a problem with altimetry data. From what I have understood from the paper, by "geolocation" you actually mean being able to tell the orientation of the iceberg, so the points over the iceberg are compared with themselves even when the iceberg has rotated.

E.g.: Say you have points **n** and **m** assigned before calving ($t0$); after $\Delta t$ amount of time drifting, CryoSat-2 measures the height indicated by the yellow "pixel". If you don't know the iceberg's orientation, you could assume the height taken refers to point n (as if the iceberg has not rotated,

transparent image in the background. But if you to know the new orientation (by "geolocating" the iceberg using imagery), you see that the height measure actually refers to point m.

[Figure]

Did I understand this correctly? Either way, I think this needs to be clarified on the text. I would say something like "determine the iceberg's orientation" instead of "geolocate" or, in case you really mean geolocate, to explain that altimetry data is not referenced to lat-lon positions.

You are correct. What you described is what we meant. We rephrased 'geolocation' to 'colocation' throughout the paper. Thanks for spotting!

Line 26: Very nice overview paragraph!

Thank you.

Line 128: *"(ii) using measurements of the semi-major and semi-minor axes provided by the NIC **and assuming an elliptical shape** and (iii) using measurements of their arc lengths recorded in satellite altimetry **and assuming a circular shape**."* Just to be bluntly clear.

Done.

Line 176: *"to align all images to a common orientation (Fig. 3)".* From my point of view, the common orientation can be seen in Figure 7a-l, not in Fig.3. Unless you are talking about the altimeter track orientation – if so, please specify.

We moved the reference of Figure 3 to another sentence and changed this reference to Figure 7a-l.

Figure 3: There is a O**k**t in figure (a) and a De**z** in figure (c). I'd also indicate which sensor each image is from, which might not be obvious to a person that does not work with satellite data.

Well spotted! We corrected the spelling mistakes and added the sensors.

Line 237: How exactly are you accounting for the iceberg drift? Have you calculated the distance the iceberg has travelled between the CryoSat and MODIS/Sentinel measurements*? And what does this imply? I assume you look at different locations between altimetry and imagery to find the iceberg (which has drifted between measurements). Or do you take the height measurements from the same points the iceberg is occupying in the imagery, assuming they didn't move but assigning an uncertainty to the drift?

If the image is from a different date than the CryoSat track, we correct the distance travelled based on the daily iceberg locations from the AIT database. In any case, we account for the drift in our

uncertainty estimate, which is higher the larger the time separation is. We added this explanation in the paper to clarify.

*In model simulations, the icebergs' average speed is 0.14 m/s, which means it would have moved around 500 m in 1h.

Then, you assume that the iceberg has kept the same orientation but account for an error of 15°/day. How do you combine all those uncertainties to get to a final error in the freeboard estimate? Could you include an equation or give more detailed information on the supplementary material?

We updated the text to make this clearer. Our drift speed of 3 km/day is based on findings by Scambos et al. (2008), who find a net movement of approximately 2–3 km/day south of 65 degrees South in free-drifting periods for two icebergs with GPS stations installed on them.

Although the methodology is, in general, very well described, I'm not sure about this part. And that makes me wonder if using the imagery to infer spatial height variability (assigning the altimetry data to specific points on the iceberg) is worth the trouble. I see that the final variability is reduced when this procedure is done (compared to the averaged freeboard estimate) on line 340, but as you mention, this is probably the case because the iceberg has a "homogeneous thickness", i.e., if you mistake one grid cell for another it is not a big deal. And, if the thickness is homogeneous, then the averaged height along the track (without "geolocation") should be good enough – which indeed is the case, since you mention that the different methodologies' results are in good agreement. So, maybe this "geolocation" is more useful when one is dealing with a non-tabular iceberg, where the thickness from point to point varies a lot and doing simply a freeboard average along the iceberg would lead to the significant loss of spatial information. However, in this case, the uncertainties brought by the time difference between the altimetry and imagery would be much more damaging to final results, since mistaking one point of the iceberg for another would imply much larger errors.

You are right. For this iceberg we find that colocation improves the uncertainty only slightly, but to generalise this finding more icebergs with different topographies would need to be studied. The Thwaites Ice Shelf is also particularly rugged and crevassed, which leads to high variations within the same grid cell. For other icebergs with more heterogeneous freeboard across the iceberg that are less crevassed, colocation might have a larger impact. We added these explanations to the text as well.

Line 253: I assume by thickness you mean draft + freeboard?

Correct. We clarified this in the paper, too.

Line 267: Define SWE.

Done.

Figure 7: The caption needs to be updated with the subplots' correct letters.
"*m)* mean difference of each new overpass along time."
"*a-l)* freeboard difference in each grid cell"

Done

Also, in a-l, the $\Delta t$ is indeed always positive? i.e., is the satellite image taken always after the CryoSat overpass? If not, you could differentiate them with a minus sign for images taken before the CryoSat overpass.

Good idea to differentiate between images taken before and after CryoSat. We updated the numbers in the figure. We also noticed that the time differences given in the figure so far were only approximate and based on the times from the CryoSat file names. In the calculation and for the new updated time differences, however, we use the time of the overpass over the iceberg, which is up to 1.5 hours later and describes the offset to the image more accurately.

Figure 8: (a), (b), (c), and (d) labels missing from plots

Done

Line 376: I'd actually move the reference to Figure 8d to the next sentence:
*"This amounts to 117 ± 38 m of thinning **(Fig 8d**)"*

Done

Line 391: Fig 8d actually shows the thickness **differences**

True. We deleted the reference here.

Line 400: "*volume changes due to fragmentation become the dominant source of ice loss towards the end of our survey".* Isn't that funny, though? I'd imagine it is much easier to break a large piece of ice than a small one. And it even somewhat contradicts what was said in line 57: "breakage dominates over melting for large icebergs". Could you offer an explanation, then, why fragmentation becomes more important at the end of this iceberg's life?

A similar behaviour has also been reported in other studies (Bouhier et al., 2018; Scambos et al., 2008). The main drivers of fragmentation are surface melting, which can lead to a rapid disintegration (Scambos et al., 2008) and wave erosion or wave stress (Wagner et al., 2014) which increase the further North (i.e. surrounded by open ocean and warmer air temperatures) the iceberg gets. Even at the end of this study period, the B30 iceberg is still considered a very large iceberg (it still has a long axis of 35 km according to the NIC). In a model used by England et al. (2020) for example icebergs break up as long as they are at least ~900 m long. We added a sentence to discuss this in the paper, too.

Line 406: A relevant reference here is Martin and Adcroft (2010) when talking about bergy bits.

Thank you. We added the reference here.

**A note about the colormaps used in the figures:** although I do enjoy rainbow colormaps, it is good to think about inclusion – namely, colorblind readers. Even for me, the colormap in Figure 1 and 2 could get confusing: it starts with red and finishes with… red. Of course, we infer which color corresponds to which year just by following the progression of the iceberg, but that should be clear from the color scheme as well. The colormap in Figure 4c and 4d is a good one: starts with bright colors, finishes with dark ones. When in doubt, there are resources such as:
https://www.color-blindness.com/coblis-color-blindness-simulator/
where you can upload you figure and see how it looks like for someone with color vision deficiency. Also, I prefer discrete color bars rather than continuous ones, so you can really see what values are attributed to each color. I don't expect you to change all your figures now, but I wanted to throw the idea out there to keep in mind for next publications.

Thank you for your thoughts and the reference. We changed the colourmaps of Figure 1 and 2.

**Technical corrections**

Line 27: Be mindful of commas. I found them missing from some places, but there could be more around. Here: "*At any time, (…)*".

Done

Line 34: Check if there is a space between *"production"* and the following parenthesis.

Done

Line 56: *"iceberg melting, to first order, (…)"* or rearrange the sentence such as *"found that iceberg melting is proportional to water temperature to first order (…)"*.

Done

Line 71: *"also included"* – just to be consistent with "*studied*" in line 70.

Done

Line 73: *"employed altimetry measurements"* – same as above.

Done

Line 125: *"The initial area"* (also, could you provide the length of the iceberg?)

Done. Yes, we added this: The iceberg's initial length was 59 km according to the National Ice Center.

Line 74: *"Bouhier et al. (2018) analysed"* – same as above; plus, that's a long sentence. I'd just split it into 3 shorter ones, one for each citation.

Done

Line 75: *"(…) geolocation. Li et al. (2018) calculated"*

Done

Line 76: *"(…) overpasses. Han et al. (2019)"*

Line 78: "*When thickness and area changes are combined, it is possible (…)*"

Done

 Line 174: *"CryoSat-2 overflights."* – the end of this sentence doesn't read well, so you can just remove it.

Done

Line 287: Maybe *"less accurate"* would sound better than "*more approximate*"

Done

Figure 6: On the caption: *"over time (a) and as scatter plot (b)."*

Done

Line 315: *"larger sections  more rapidly."*

Done

Line 320: *"In 2018, (…)"*

Done

Line 341: Do you mean Figure 7m instead of 5a?

Yes, corrected.

Line 393: *"To compute changes in mass, we (…)"*

Done

Line 412: *"thickness, and volume (…)"*

Done

**Reviewer #2 (Jessica Scheick), 13 April 2021:**

**General Comments:**

The manuscript tracks Antarctic iceberg B30 using satellite imagery and altimetry observations. The authors utilize these observations to determine changes in iceberg area, freeboard, and volume through the iceberg's drift from its calving location from the Thwaites Ice Shelf. Their analysis investigates the viability of using semi-automated methods for estimating iceberg area, the importance of geolocation in estimating iceberg freeboard, and the impact of including snow accumulation and snow and ice density variations in computations of ice thickness.

This work contributes to our ability to scale up analyses of iceberg drift and disintegration by quantifying the limitations of some common assumptions and uncertainties of various methods, especially as they relate to semi-automating the analysis. However, it would benefit from a more thorough exploration of these assumptions and discussion of where future work should focus on improvements. A few key areas of focus for improvement prior to publication are:

Thank you for your detailed review and the time you put into it. We responded to all of your comments below and hope this clarifies our work further and resolves the misunderstandings.

1. Overall clarify and improve the motivations and contextualization within the literature. What is unique about this investigation? The novel contributions of this work (investigating the influence of geolocation on iceberg freeboard estimates; including snow accumulation and density variations in thickness calculations) only become clear towards the end of the manuscript. Many areas of the methodology and discussion are lacking citations and explicit connections between previous results and this investigation (a few specific cases are pointed out in the line comments, below, but this list is not exhaustive). Why did you choose to focus on iceberg B30? What should we take away from this investigation, and how should it inform our future work? What are the critical next steps needed to further this work?

   This is quite a challenging and dispiriting opening series of comments. The main novel contributions of our study are all mentioned in the abstract; not overstating their novelty explicitly is our preferred writing style for a study of this nature. We have though tweaked the paper title to be a little more emphatic, we have added reference to our use of meteorological data in the abstract, and we have added a few sentences at the end of the introduction to restate the novel contributions and to explain our choice of B30 as a test case and the significance of the berg itself. We added 17 additional references and also cited more existing references in the methods and discussions. In total, we now cite 76 papers,

which is close to the upper limit (80) for the journal and significantly higher than the upper limit for many other journals, and we have provided specific responses to the related comments. Finally, the take-away messages and future work are already mentioned in our conclusions. We note that none of these concerns relate to the scientific contribution of our study, which leads us to believe that we may have misunderstood their intent.

2. Clean up precision of language, passive voice, and extraneous phrases ("more recently, for example"). This includes separating run-on sentences and connecting ideas throughout and between paragraphs (there are a few abrupt transitions and locations where critical information is presented a page or two later than the reader needs the information).

   We corrected the language and the abrupt transition.

3. Closely examine the text for statements that need further quantification, explanation, citation, etc. This is similar to 1 and 2, but refers to particular statements like "While manual delineation provides the most consistent and accurate area estimate" or "boundary detection techniques" or "large" icebergs or iceberg "area" and "thickness". Additional details on your approach, methods, and definitions will convince the reader they agree with your interpretations and make your method reproducible.

   We added more details and references to the methods, explicit definitions for area and thickness, and used the word 'significantly' more carefully. Please also see our responses to the related specific comments.

4. Data and code access. The manuscript does a reasonable job of outlining what computational tools are used but does not provide enough details to make the study reproducible nor indicate where readers can get more information. What software and data versions are you using? What corrections did you apply? Is your code publicly available? Why or why not? Are the iceberg polygons available?

   Thank you for spotting this. We added code and data availability statements. The corrections applied, software and data products are mentioned in the methods section (e.g. lines 121, 135-140, 186, 188, 191-192, 197, 199-201, 220-229, 237-238, 263-266, 270, 281-282) and acknowledgement (lines 242-244). For the AIT database we added the version number and for the CryoSat-2 data we added the baseline. For MODIS, Sentinel-1 and the ERA-5 data there is only one version available and no specific version numbers are given in the archives.

**Specific ("line") comments:**

*Abstract:*

p1 Line 15: You're comparing a time series to a geospatial track?

We clarified this in the paper.

p1 Line 18: geolocation of imagery reduces the uncertainty of what by 1.6 m? Iceberg location? Freeboard?

Freeboard. We clarified the sentence.

Introduction:

p2 Line 52: "ice shelf barriers" = "ice shelf fronts"? I think this may be a British/American English difference, since I'd previously only heard this term in reference to Ross Ice Shelf

The term "ice shelf barrier" refers to the role of the ice shelf as an interface between the ice sheet and the ocean and not just the specific calving front (although of course all barriers have a "front"). For a recent discussion see the reference cited (Shepherd et al., 2019).

p2-3 Lines 55-56: be careful not to mix terms: melting and breakage are both forms of mass loss

We rearranged the sentence to avoid misunderstandings.

p3 Line 65: Explicitly state your focus on tabular Antarctic icebergs, versus icebergs generally

We modified this sentence and made our focus on tabular icebergs clearer throughout the paper including a modification of the title.

p3 Line 83: the studies cited here occurred before the ones cited in the previous sentence…

We changed the sentence.

p3 Line 85: this is an abrupt transition. Also, is your method less labor intensive?

We added a paragraph to mitigate the abrupt transition. In our experience, producing elevation data from stereo-photogrammetry and interferometry are certainly labour intensive by comparison to satellite altimetry. The text was updated to clarify.

Iceberg location:

p4 Line 103: longer than 6 km in what dimension (their longest? How is this estimated?)

6 km refers to the long axis. We believe that they track icebergs, which are visible in their scatterometry data. For more details please see Budge & Long (2018) and Stuart & Long (2011). We also added the latter reference to the paper.

Initial iceberg shape, size and calving position:

p6 Line 126: The initial area may be more appropriately reported in the next subsection.

As the initial shape and area are derived from the same polygon, we decided to state them in the same subsection.

Iceberg area:

p6 Line 128: please clearly define "iceberg area". I am assuming for the purposes of this review that "area" refers to the two-dimensional, plan-view, non-submerged portion of the iceberg

Your assumption is correct. We also clarified this in the paper.

p6 Line 130: I would like to see some justification for the statement that manual delineation provides the most consistent and accurate area estimates. From my experience, selecting consistent iceberg boundaries manually is non-trivial and can result in multiple "correct" delineations with vastly different areas. The introduction of multiple operators can further increase the spread of possible surface area estimates.

In our results section and Figure 6 we show that manual delineations are much more consistent and realistic than the other approaches. As you say even manual delineations come with an uncertainty. We account for this in our uncertainty estimates.

p6 Line 137: Please include which orbital and radiometric corrections you're using.

We apply precise orbit files and the radiometric corrections provided by the calibration Look Up Tables in the Level-1 products using the ESA SNAP software. For more information please see the hand book or help of SNAP, which is an open source software.

p6 Line 144: What boundary detection techniques do you use? How do you select any parameters used in these techniques, and how do these affect your area estimates?

We use matlab's bwboundaries function, which implements the Moore-Neighbor tracing algorithm modified by Jacob's stopping criteria. As we only use a limited number of area outlines, we haven't experimented with different techniques or parameters and simply use manual delineations when this method fails. We added more details in the paper, too.

p6 Line 146: What "rules" (explicit or implicit) are you using during manual delineation? Shadows cast on the sea ice? Texture differences? How do you handle areas that are "blurry"?

If areas are blurry due to cloud cover, we use multiple MODIS images together, where different parts are covered by clouds. Mostly the colour (or grey scale backscatter for Sentinel 1) is enough, but in some MODIS scenes where the iceberg is surrounded by sea ice, you can also detect a texture difference when zooming in (e.g. Fig. 3b). We also clarified this in the paper.

p6 Line 147: What shape kernel do you use for the shrinking and expansion?

We did not use a kernel, but simply moved each polygon point by one pixel (lines 146-147).

p6 Line 148: What is the standard deviation on this mean relative difference?

We added the standard deviation of 0.9 % to the paper.

p7 Line 157: How are the NIC axes determined? If this is done manually (as stated in line 160), then you cannot argue this approach is less time consuming or subject to individual judgement (line 156).

NIC axes are indeed derived manually, but we think it's much faster to measure two axes than clicking the whole outline ('delineation of their full perimeter', line 154). Furthermore, this is an operational service (line 158) and therefore it saves time for end users, who don't have to duplicate their work.

p7 Line 163: Can you compare one of your area estimates to one of the elliptical ones from NIC and combine the area datasets?

We compare our estimates to the estimates of semi axes by NIC assuming an elliptical shape in Figure 6 and section 3.1. A direct comparison is shown in the scatter plot (Fig. 6b).

p7 Line 166: How are iceberg arc lengths determined from CRYOSat-2 data? Is this an existing product or are you deriving the iceberg arc lengths?

We are deriving the arc length from the same tracks that we use to estimate the freeboard and thickness. This was clarified in the paper.

p7 Line 168: Please provide additional information on the "significant variations" in area estimates from your third method.

We rephrased 'significant' to 'considerable'. What we mean is that the area estimates based on one arc length are not consistent and vary depending on where the CryoSat track crosses the iceberg.

p8 Line 169: what dimension is the moving mean computed across?

It's computed over time and this was added to the paper.

Iceberg orientation:

p8 Line175: is the rotation performed manually or automatically?

The rotation angle is adjusted manually.

Initial iceberg freeboard:

p9 Line 198-201: Were the outliers removed sequentially using the filters described (i.e., were the median and mean filters applied to the range of freeboard heights subsequent to the removal of values outside the 20-60 m range?). Also, with median it is customary to use standard absolute deviation, rather than standard deviation. What criteria were used to select the 20-60 m initial filter, and could similar removal of outliers be accomplished with only one or two median or mean filters?

The filters were applied successively. 20-60 m are based on the findings by Tournadre et al. (2015) and ensure that all remaining measurements are actually from the iceberg (lines 223-224). This step is important, because including returns from e.g. nearby land, sea ice or simply outliers could bias the mean and median. We added the reference here, too.

p9 Line 202: What criteria are used to detect and exclude crevasses?

The filters described above are used to exclude crevasses. We clarified this in the paper.

p9 Line 209: Be aware of using "significantly" without quantification.

We rephrased "significantly" to "considerably".

Iceberg freeboard change:

p10 Line 220+: It's unclear exactly what filtering is done here to extract icebergs. Is land excluded geospatially, or is it excluded using one of the height filters? It might help the reader to be explicit

that you are automatically extracting iceberg freeboards from tracks, motivating the need for multiple filtering steps.

Yes, we exclude land using these height filters, because land masks are not accurate enough along advancing and retreating ice-shelves and the iceberg is drifting very close to the ice-shelf margins occasionally. We also added a sentence to point out that we are automatically extracting the iceberg from the tracks with these steps.

p10 Line 226: The editing steps to remove rugged features and crevasses are not previously described (as such).

They are described in lines 193-201.

p11 Line 248: If you were to compute an iceberg freeboard for the pre-calved iceberg using just one of the tracks used in your compilation, how would that value compare to the mean freeboard calculated using the composite? Making this comparison would be a compelling way to show that your mean along-track freeboard computations can be directly compared to the mean surface height prior to calving as a measure of changes in freeboard.

This is indeed a nice idea for validation! We made this calculation for each of the 15 tracks over the pre-calved iceberg and find a standard deviation of 2.8 m compared to the mean initial height of 49.0 using all the tracks. This was also added to the paper.

Iceberg thickness:

The equations presented in this section are described in the text but are not incorporated into it. Instead, they hang between paragraphs. Constants used within the equations should be explained and/or cited.

We double-checked and couldn't find any constant that was not explained. We cited additional references for the uncertainty values.

Results and Discussion:

p16 Line 343-345: The reader could benefit from this information on the variability of freeboards being presented earlier in the manuscript.

We also presented this information in line 206.

Iceberg thickness change:

p17-18: this section presents a lot of critical information but is rather confusing because it switches between negligible and non-negligible influences on iceberg thickness and density. Please revise to flow logically and indicate which processes were considered and which were included in the final calculations.

We arranged this section based on the different parameters, therefore e.g. firn densification comes after ice density changes and snow melting comes after snow accumulation.
All processes that are mentioned in the methods section were included in the final calculation.
The other processes with negligible influence are only shortly mentioned here as a discussion. We

made it more explicit in the paper that the effect of surface melting is not applied. For firn densification this is already stated (line 365).

p18 Line 366: Is the snow density averaged vertically, assuming a uniform horizontal thickness?

We directly calculate the mean snow density; it's not explicitly averaged vertically (Line 255: $\rho_s$ is the column-average snow density). Only for the ice density we model the density profile.

p18 Line 369: where were above-freezing degree hours calculated?

We mention how it's calculated in this line. The data (2m air temperature data) was introduced in section 2.7. Or do you mean where geographically? It's calculated along the iceberg's trajectory.

p19 Line 377: Is a linear, annual average the most representative of iceberg processes?

Icebergs are not melting linearly and at a constant rate, but this allows us to compare our work to previous studies and set the findings into context (line 377 onwards).

p20 circa Line 385: The authors clearly articulate and demonstrate the importance of including snow accumulation in thickness computations. It would be great to see some further discussion of this. Some potential avenues for exploration include noting over what temporal and/or spatial scales (e.g. when the iceberg is close to the coast) these effects are important, recommendations for when this is a critical component that should be included in estimating iceberg thickness, and or discussing the limits of including snow accumulation but excluding wind scour and other snow removal processes.

Figure 8 suggests more or less linear snow accumulation. The influence of the snow layer is probably larger when the iceberg is melting less and becomes larger the longer the iceberg survives. For a real judgement, more icebergs in different environments need to be accessed, though. We added a short discussion on this in the paper.

Iceberg volume and mass change:

p20 Line 389-390: This line is a great example of a clear, simple statement that provides information about your logic/assumptions and objectives. Awesome!

Thank you.

p20: What are the implications of your assumptions about shape on volume? This is a critical assumption in estimating iceberg volume that has been quantified (to the best of our ability) and discussed in the literature (e.g. Enderlin and Hamilton 2014, Sulak et al 2017, Schild et al 2021, and many others). Your discussion needs to address the assumptions you've made and justify the interpretation on the results accordingly.

Iceberg shape plays a role for small icebergs. However, large tabular icebergs like B30 inherit their shape from their parent ice shelves and therefore have rather homogenous thickness and near vertical walls . So, they are treated as prisms with constant area from the surface to the base. B30 for example was 59 000 m long and 315 m thick at calving and had a length to thickness ratio of 187:1. Slightly tilted sides have minor effects on the total volume and we anticipate that the deviations from vertical occur in both directions, so they approximately even out across the whole side surface (e.g. Orheim, 1987). The papers you mention all refer to small icebergs with totally different dimensions and are not transferable to large tabular icebergs.

Since you brought this up, we, however, did a few quick calculations using the formulas from Enderlin and Hamilton 2014 and simple trigonometry (see below). These show that the B30 iceberg would need near horizontal side walls (tilted by 1.5 degrees compared to horizontal) in order to have a conical shape. This seems very unrealistic.

If all sides were tilted by 5 degrees compared to vertical this would lead to a 0.1 % difference in volume and an absolute difference of 0.6 km^3 compared to an uncertainty of 57 km^3 (line 398). These results confirm that deviations from our prism assumption with constant area have negligible impact on the volume of this tabular iceberg. In the paper we made our focus on large tabular icebergs and their main differences to small icebergs clearer, adding the word 'tabular' to the title, and explanations to each chapter. We also added a short discussion on iceberg shape.

[Figure]

Half the iceberg's length = 29 500 m

$\alpha$

Draft for cylinder shape = 798 m

→ $\alpha = 1.5°$

$G_1 = 1500\ km^2\ ..iceberg\ area$

$T = 0.315\ km\ ..iceberg\ thickness$

$\alpha = 85°$

$r_1 = \sqrt{G_1/\pi}$

$h_1 = r_1 \cdot \tan(\alpha)$

$h_2 = h_1 - T$

$r_2 = h_2/h_1 \cdot r_1$

$G_2 = r_2^2 \cdot \pi$

$Volume = \frac{1}{3} \cdot (G_1 \cdot h_1 - G_2 \cdot h_2) = 471.9\ km^2$

$Volume\_cyl = \frac{1}{3} \cdot (G_1 \cdot T) = 472.5\ km^2$

$Difference = 0.1\ \%$

p20 Line 407: Do you calculate freshwater flux? If not, why?

We can only give a lower bound estimate of the freshwater flux, which is the volume loss due to basal melting of the mother iceberg. As the fragments, which are lost from the sides, are not tracked by the database and most of them are too small to be tracked by altimetry, we can't say how quickly they melt and therefore how much they (and hence fragmentation) adds to freshwater flux. This is also discussed in the paper (Lines 404-407). We added further references on this topic in the paper to point readers, who are interested specifically in the fresh water flux, to the related literature.

Figures (overall):

What uncertainties are shown (two sigma?)

The uncertainty of freeboard is one standard deviation. This was added to the figure caption. The other uncertainties shown are estimated as described in the methods.

Figure 1: Why is there such a large data gap circa 2018-2019?

It's a data gap in the Antarctic Iceberg Tracking database.

Figure 2:It would be helpful to have the icebergs in panel (b) oriented as they are in panel (a). Is the orientation of all of the depictions in panel (b) with north to the left? What is the influence of outline complexity on area? there is a very clear difference between the level of detail in the iceberg outlines that were manually derived versus the edge detection derived outlines.

The icebergs in panel b are depicted in a polar stereographic projection (information added to the caption). We didn't investigate the influence of outline complexity on area systematically, but there were a few instances where we first clicked the outline manually and then discovered the automatic technique and we remember that the difference was small compared to the total area.

Figure 3: Can you really see the iceberg orientation in panel (l)? Also, why doesn't the initial iceberg fully encompass the iceberg depicted in later panels? This suggests to me either the initial iceberg outline needs to be modified or the rotational alignment should be improved.

For example in panel b the iceberg has sea ice frozen to its side. When you zoom in, the sea ice is slightly transparent/darker and has a different texture than the iceberg. Using a sequence of several images helps to identify this. In panel l zooming in and using several images in combination, where the clouds cover different parts of the iceberg helped to identify it. This is definitely our worst quality image, though. The following image was taken only one day later and really helped to find the iceberg in the image from the previous day. We clarified this in section 2.3 in the paper.

[Figure]

Figure 4: It's interesting that one of the areas with the largest number of observations is also one of the areas with a comparatively higher standard deviation. Is this a particularly complex or crevassed region?

We can't judge, but this is indeed what the figure suggests.

Figure 5: This figure is immensely helpful for visualizing your workflow. It might be helpful to the reader to isolate some of the filtering steps to illustrate why each one is needed (perhaps as a supplementary figure?).

The figure separates the main two steps: 1. Identifying where the iceberg is (blue) within each track (black) and 2. Which parts of the iceberg (blue) are crevasses (remaining heights in red).

Figure 7:Why does the difference in freeboard have such a potentially large positive range (it appears there are no values larger than about 7 m)? Plots are mislabeled relative to the caption. What is the shaded region showing? Label the axes in plot n. How do you explain the large variability (specifically the increases in freeboard)?

Do you mean why our y-axis ends at +15 m? This is to show the whole uncertainty range (shaded region), to leave room for the legend and because we wanted to have a y-axis label at the upper end (labels every 5 m). Mislabeling was corrected. Thanks for spotting this.

The increase in freeboard and thickness is indeed very interesting. We added a paragraph and further references to discuss the possible causes. For more details, please see our response to the short comment from Xuying Liu below and the tracked changes.

Grammar:

- Heading titles have inconsistent capitalization

  Corrected

- inconsistent use of Oxford comma

  Corrected

- watch for possessive apostrophes (both missing and extraneous)

  Corrected

- p16 Line 341: there is no (a) in figure 5

  Corrected (see reviewer 1)

**Short comment (Xuying Liu), 5 April 2021:**

Thank you for your work on iceberg thickness and volume changes of B30.

I've got a question while reading. Figure 8(d) shows a significant trend of thickness descending from 2012 to 2018. But there's also some rises of thickness shown by the figure. What may cause the iceberg being thicker during its drift? It would be appreciated if you could share your opinions.

We added the following paragraph to discuss this in the paper:

"Besides the observed thinning, the iceberg also seems to slightly thicken between mid-2014 and early 2015. During this time B30 was very close to the coast (Fig. 3b-d). Therefore, a range of processes – both physical processes that impact the actual thickness of the iceberg and processes that impact the freeboard measurement – could have caused this gain in thickness: First of all, iceberg thickness can increase through marine ice formation, when the iceberg is surrounded by very cold water. (Little et al., 2008) found freezing beneath ice-shelves concentrated along their western side and B30 was indeed located at the western side of Getz ice shelf at this time (Fig. 1, 3b, c). It can also grow through snow accumulation on the surface, which we account for, but only based on reanalysis data and there might be additional local snowfall or snow accumulation through strong katabatic winds from the near-by continent (Fedotov et al., 1998). Furthermore, external forcing from collisions with the adjacent ice-shelf might have led to a deformation (MacAyeal et al., 2008) and hence a compression in some parts. All of these processes can cause a physical increase in iceberg thickness. Apart from that, a short (partial) grounding could lead to higher measured iceberg freeboards (Li et al., 2018). Also surface melting could shift the scattering horizon of CryoSat-2

(Otosaka et al., 2020) and therefore appear like a freeboard increase. Indeed we observe a steep increase in degree hours around the turn of the year 2015. What caused the signal in this instance is hard to disentangle. Most probably, it was a combination of several of the mentioned effects."

---

## Author Response (AR2)

Response to Jessica Scheick:

Thank you very much for your thorough reading and assessment of this manuscript. We adopted most (15) of your suggestions for the final polish and explained the remaining 5.

General Comments:

The authors have done a great job addressing the comments from both reviewers and the online discussion. Issues with clarity in the first draft, including misused and ambiguous terms and inexplicit assumptions, were addressed. The investigation and methodology are now understandable to readers unfamiliar with the project (the questions contained in the overview comments were intended to summarize the main ideas of some of the line corrections and provide a set of key motivating questions to keep in mind during revisions from the perspective of someone unfamiliar with the study, not discredit the scientific merit). The additional discussion elements and improved clarity effectively connect previous results to this investigation, making this a robust contribution to the iceberg literature. Several added details greatly contribute to the study's reproducibility, though I would encourage the authors to consider adopting FAIR (Findable, Accessible, Interoperable, Reusable) practices for their code and data. With a few additional technical clarifications and corrections, this manuscript will be ready for publication.

The use of Oxford commas is still inconsistent (it appears that they're not used in lists of "items" (e.g. size, shape and area) but are used for lists of "ideas" (e.g. a said x, b said y, and c said z). Line 471 is different between the track changes and revised versions, which I did not compare in other cases and may explain the inconsistencies throughout.

Oxford commas should be consistent now.

Specific Line comments:
Line 34: add "influence" or "enhance" before "melting" to keep the list consistent.

Done

Line 59: remove "for example"

Done

Line 89+: Excellent – This paragraph now clearly captures the motivations for your investigation and sets the tone for the work presented in the rest of the manuscript.

Thank you.

Line 141: images, not imagery

Done

Line 145: Perhaps "… to apply the orbital and radiometric corrections provided with the imagery." This will convey to the reader the information provided in the author comments.

Done

Line 157: "To estimate the uncertainty of our delineations, we buffer the polygons by the source imagery pixel width… and calculate the resulting differences in area." It sounds like a vector operation was applied (buffering), but the current phrasing suggests a raster operation (shrink/expand by pixels), which would require use (perhaps implicitly by the software) of a kernel.

Done

Line 159: uncertainties are missing units

The units are % in both cases.

Figure 2: is north up or to the left in panel b?

Panel b is in polar stereographic projection (with 0 degrees longitude up).

Line 188: was there a particular temporal threshold you used (perhaps I missed it in the text earlier?).

Yes, we mention it in lines 252-253.

Line 215: I'd suggest making this even more explicit: "When outliers and crevasses are excluded…"

We added this.

Line 250: remove quotes ("colocated")

Done

Line 258: remove l (collocated)

Done

Fig. 3: could you use one of the clearer images as a background?

The clearer images have a longer temporal offset to the altimetry data and therefore we used the image shown in Figure 3 for colocation, but the clear images helped to identify the shape of the iceberg in this image.

Line 315: remove "large"

Done

Line 317: I suggest "manually delineated icebergs" or "deriving iceberg outlines". Delineation implies outlines.

Done

Line 353: pre- and post-calving might be clearer terminology than "new overpasses".

Done

Line 376: consider using parentheses instead of dashes so it's easy to tell at a glance that 3.3 is positive and not negative.

Done

Line 391: Do you mean the "mean change in height due to firn densification is…"? Otherwise, the units should be kg/m3 per year to represent a change in density.

Done

Figure 7: The scale bar for panels a through l greatly exceeds the actual range of the data. The y axis in panel m is scaled appropriately, as the authors note.

We chose the scale bar for a-l to be symmetrical around zero, as it is a common choice to plot zero change white and positive or negative changes in red and blue respectively.